# Loophole-free Bell inequality violation with superconducting circuits

Simon Storz[1✉], Josua Schär[1], Anatoly Kulikov[1], Paul Magnard[1,10], Philipp Kurpiers[1,11], Janis Lütolf[1], Theo Walter[1], Adrian Copetudo[1,12], Kevin Reuer[1], Abdulkadir Akin[1], Jean-Claude Besse[1], Mihai Gabureac[1], Graham J. Norris[1], Andrés Rosario[1], Ferran Martin[2], José Martinez[2], Waldimar Amaya[2], Morgan W. Mitchell[3,4], Carlos Abellan[2], Jean-Daniel Bancal[5], Nicolas Sangouard[5], Baptiste Royer[6,7], Alexandre Blais[7,8] & Andreas Wallraff[1,9✉]

Superposition, entanglement and non-locality constitute fundamental features of quantum physics. The fact that quantum physics does not follow the principle of local causality[1–3] can be experimentally demonstrated in Bell tests[4] performed on pairs of spatially separated, entangled quantum systems. Although Bell tests, which are widely regarded as a litmus test of quantum physics, have been explored using a broad range of quantum systems over the past 50 years, only relatively recently have experiments free of so-called loopholes[5] succeeded. Such experiments have been performed with spins in nitrogen–vacancy centres[6], optical photons[7–9] and neutral atoms[10]. Here we demonstrate a loophole-free violation of Bell's inequality with superconducting circuits, which are a prime contender for realizing quantum computing technology[11]. To evaluate a Clauser–Horne–Shimony–Holt-type Bell inequality[4], we deterministically entangle a pair of qubits[12] and perform fast and high-fidelity measurements[13] along randomly chosen bases on the qubits connected through a cryogenic link[14] spanning a distance of 30 metres. Evaluating more than 1 million experimental trials, we find an average $S$ value of $2.0747 \pm 0.0033$, violating Bell's inequality with a $P$ value smaller than $10^{-108}$. Our work demonstrates that non-locality is a viable new resource in quantum information technology realized with superconducting circuits with potential applications in quantum communication, quantum computing and fundamental physics[15].

One of the astounding features of quantum physics is that it contradicts our common intuitive understanding of nature following the principle of local causality[1]. This concept derives from the expectation that the causes of an event are to be found in its neighbourhood (see Supplementary Information section I for a discussion). In 1964, John Stewart Bell proposed an experiment, now known as a Bell test, to empirically demonstrate that theories satisfying the principle of local causality do not describe the properties of a pair of entangled quantum systems[2,3].

In a Bell test[4], two distinct parties A and B each hold one part of an entangled quantum system, for example, one of two qubits. Each party then chooses one of two possible measurements to perform on their qubit, and records the binary measurement outcome. The parties repeat the process many times to accumulate statistics, and evaluate a Bell inequality[2,4] using the measurement choices and recorded results. Systems governed by local hidden variable models are expected to obey the inequality whereas quantum systems can violate it. The two underlying assumptions in the derivation of Bell's inequality are locality, the concept that the measurement outcome at the location of party A

cannot depend on information available at the location of party B and vice versa, and measurement independence, the idea that the choice between the two possible measurements is statistically independent from any hidden variables.

A decade after Bell's proposal, the first pioneering experimental Bell tests were successful[16,17]. However, these early experiments relied on additional assumptions[18], creating loopholes in the conclusions drawn from the experiments. In the following decades, experiments relying on fewer and fewer assumptions were performed[19–21], until loophole-free Bell inequality violations, which close all major loopholes simultaneously, were demonstrated in 2015 and the following years[6–10]; see ref. 22 for a discussion.

In the development of quantum information science, it became clear that Bell tests relying on a minimum number of assumptions are not only of interest for testing fundamental physics but also serve as a key resource in quantum information processing protocols. Observing a violation of Bell's inequality indicates that the system possesses non-classical correlations, and asserts that the potentially unknown

[1]Department of Physics, ETH Zurich, Zurich, Switzerland. [2]Quside Technologies S.L., Castelldefels, Spain. [3]ICFO - Institut de Ciencies Fotoniques, The Barcelona Institute of Science and Technology, Castelldefels (Barcelona), Spain. [4]ICREA - Institució Catalana de Recerca i Estudis Avançats, Barcelona, Spain. [5]Institute of Theoretical Physics, University of Paris-Saclay, CEA, CNRS, Gif-sur-Yvette, France. [6]Department of Physics, Yale University, New Haven, CT, USA. [7]Institut quantique and Département de Physique, Université de Sherbrooke, Sherbrooke, Québec, Canada. [8]Canadian Institute for Advanced Research, Toronto, Ontario, Canada. [9]Quantum Center, ETH Zurich, Zurich, Switzerland. [10]Present address: Alice and Bob, Paris, France. [11]Present address: Rohde and Schwarz, Munich, Germany. [12]Present address: Centre for Quantum Technologies, National University of Singapore, Singapore, Singapore. ✉e-mail: simon.storz@phys.ethz.ch; andreas.wallraff@phys.ethz.ch

quantum state has a certain degree of entanglement and purity. This assessment, based on the observed correlations between the chosen input (the choice of measurement basis) and recorded output values (the measurement outcome) of the test, does not rely on knowledge of the inner workings of the system: a property known as device independence[23]. This allows identifying quantum states and measurements[24], certifying the correct functioning of quantum computing devices[25], and establishing common and secret keys between two parties with only limited assumptions about the used devices[26]. Further applications of Bell tests include device-independent randomness generation and expansion, extending a given random bit string in a certified manner[27,28] and randomness amplification, improving the quality of a source of randomness in a certified manner[29,30], which is a task impossible to achieve by purely classical means.

Deploying non-locality as a new resource in the context of superconducting circuits enables new applications in a system that is well set for creating large-scale quantum computers[11,31] and provides quantum communication capabilities. In addition, non-local Bell tests with superconducting circuits are unique as a macroscopic quantum system[32–34] is used, which is controlled, entangled and read out exclusively using microwave frequency radiation rather than optical frequency fields.

With superconducting circuits, Bell tests were performed that closed the fair-sampling (or detection) loophole[35], supported the assumption of measurement independence with human choices[36], and used qubits connected by an on-chip 78-cm-long transmission line[37]. Whereas these experiments all relied on additional assumptions, in this work, we set out to demonstrate a loophole-free violation of Bell's inequality using superconducting circuits. The Methods section provides a very brief introduction to the basic properties of superconducting qubits.

Addressing the locality loophole[5] (Supplementary Information section I) in a Bell test with superconducting circuits, typically housed in their individual cryogenic systems, is particularly challenging, as it requires to entangle a pair of qubits located at two sites A and B separated by a large physical distance $d$ with high concurrence $\mathcal{C}$ of the entangled state, where $\mathcal{C}$ (refs. 38,39) is a measure of the degree of entanglement present in the system. An individual trial of a Bell test begins at time $t_\star = 0$ with the choice of a pair of input bits $(a, b)$, which determine the basis in which the quantum state of each of the two entangled qubits is read out (Fig. 1). To support the assumption of measurement independence, local basis choices are realized using random number generators (RNGs). If the sites A and B are separated from each other by a sufficiently large distance $d$, the exchange of information between A and B, occurring at most at the speed of light $c$, is prohibited for times $t < t_d = d/c$ according to the laws of special relativity. If the measurement outcomes are obtained during this time interval, the spatial separation thus ensures that the chosen measurement bases and the corresponding measurement outcomes by the party at one site are unknown to the party at the other site, thereby closing the locality loophole.

For each trial of the Bell test, a high-fidelity measurement of the quantum state of the qubits at A and B, which is designed to terminate at time $t < t_d$, is performed. The readout of the qubits results in outcomes $x$ and $y$ taking on values of +1 or −1 if the qubit is detected in the ground $|g\rangle$ or excited state $|e\rangle$, respectively. Including each and every measurement outcome in the analysis of the Bell test closes the fair-sampling loophole[40,41] (Supplementary Information section I). Furthermore, the memory loophole is closed by statistically analysing the input and output data without assuming that individual trials of the Bell test are independent and identically distributed[18].

To evaluate the result of a Bell test performed in this manner, the averages of the product of the individual measurement outcomes $\langle xy\rangle_{(a,b)}$ at sites A and B are calculated to determine the Clauser–Horne–Shimony–Holt (CHSH) value[4] $S = \langle xy\rangle_{(0,0)} - \langle xy\rangle_{(0,1)} + \langle xy\rangle_{(1,0)} + \langle xy\rangle_{(1,1)}$ given the four possible combinations of measurement basis choices $(a, b)$. If the properties of the system were described by a local hidden

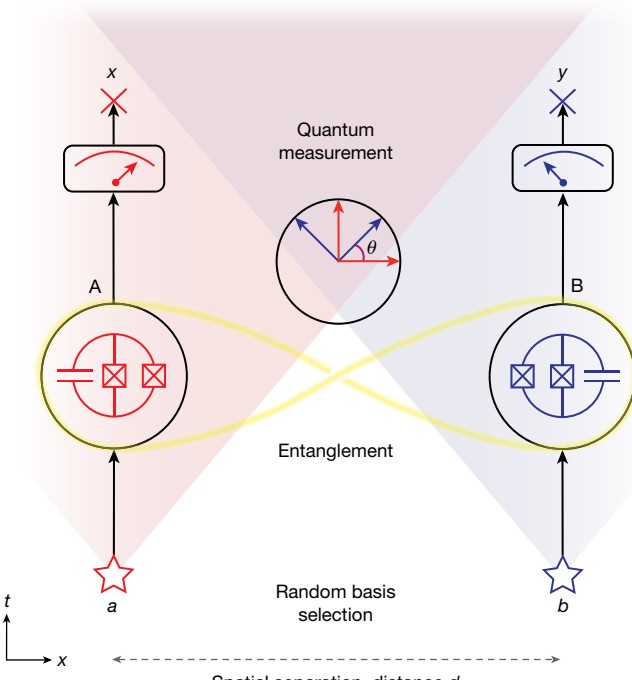

**Fig. 1 | Schematic of the Bell test experiment.** Two parties A and B choose random-input bits $(a, b)$ at the space–time locations indicated by stars and perform measurements on a pair of entangled quantum systems (in this work, superconducting circuit qubits) yielding output bits $(x, y)$ at space–time locations indicated by crosses. The shaded areas indicate the forwards light cones originating at the space–time location of the random-input-bit-generation events. The inset in the middle indicates the offset angle $\theta$ between the measurement bases of the two qubits (main text).

variable model[2], one would find $|S| \le 2$, whereas any value larger than two indicates a violation of Bell's inequality. The maximum value of $|S|$ allowed by quantum physics is $2\sqrt{2}$.

In the following, we discuss how we fulfil the requirements outlined here for realizing a Bell test with superconducting circuits closing the locality, fair-sampling and memory loopholes and supporting measurement independence all at the same time.

## Requirements

In a Bell test using an entangled pair of qubits, the degree to which Bell's inequality can be violated depends on the concurrence $\mathcal{C}$ of the entangled state and the individual qubit-readout fidelity $\mathcal{F}_r^{(A,B)}$. Together, these quantities constrain the maximally achievable Bell parameter to[42]

$$S^{\max} = 2\sqrt{2}\,\mathcal{F}_r^2\mathcal{C}(\rho_{AB}). \tag{1}$$

Thus, the CHSH inequality can only be violated if the average readout fidelity $\mathcal{F}_r = \sqrt{\mathcal{F}_r^A \mathcal{F}_r^B}$ exceeds roughly 84% and the concurrence $\mathcal{C}$ exceeds roughly 0.7, so that $S^{\max} > 2$, as shown in the contour plot in Fig. 2a.

Addressing the above requirement, previous experiments achieved remote entanglement of superconducting qubits with sufficiently large concurrence in a single dilution refrigerator[12,37,43,44] and in two refrigerators connected across a distance of 5 m using a cryogenic link[14]. In the experiments we present here, we create entanglement over much larger linear distances. In addition, single-shot readout of superconducting qubits was demonstrated in an integration time of 50 ns with fidelity $\mathcal{F}_r$ exceeding 98% (ref. 13). In a Bell test that closes the locality loophole, minimizing the duration of the readout reduces the distance $d$ required between the two parties to provide space-like separation.

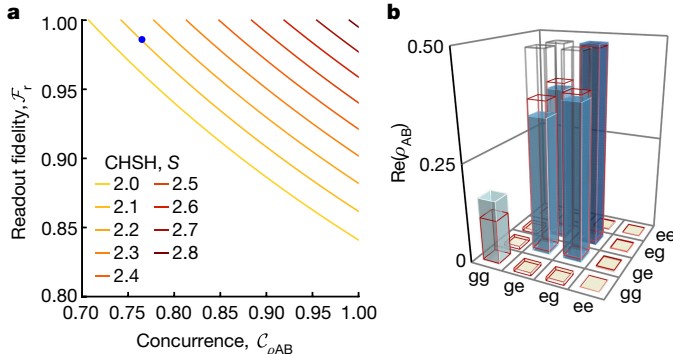

**a** 

Readout fidelity, $\mathcal{F}_r$

CHSH, $S$
- 2.0 — 2.5
- 2.1 — 2.6
- 2.2 — 2.7
- 2.3 — 2.8
- 2.4

Concurrence, $\mathcal{C}_{\rho AB}$

**b**

$\mathrm{Re}(\rho_{AB})$

**Fig. 2 | S value, entangled state and readout fidelity. a**, Calculated $S$ value for a Bell test performed in the $xy$ basis of the Bloch sphere versus (readout-corrected) Bell state concurrence $\mathcal{C}(\rho_{AB})$ and average qubit-readout fidelity $\mathcal{F}_r = \sqrt{\mathcal{F}_r^A \mathcal{F}_r^B}$. The blue data point indicates the experimentally achieved readout fidelity and concurrence (with correction for readout errors). **b**, Real part of the density matrix $\rho$ of the Bell state $|\psi^+\rangle$ reconstructed using quantum state tomography corrected for readout errors. The blue bars indicate the measured, the grey wireframes the ideal values and the red wireframes the results of a master equation simulation.

Given the expected readout time of roughly 50 ns, the time required for choosing the measurement bases at random (roughly 29 ns, Supplementary Information section II), and accounting for a margin for signal propagation times, we chose to build a cryogenic system housing superconducting circuits at a linear physical distance of roughly $d = 30$ m (Fig. 3). This provides a time budget $t_d$ in excess of 100 ns for the Bell test.

In our experiments, a pair of dilution refrigerators, one at site A and one at B, each house a superconducting qubit with circuitry for local readout and remote entanglement[12–14] cooled down to about 15 mK. In a unique set-up (Fig. 3), we connect the two circuits to each other over a distance of 30 m using a cryogenic quantum microwave channel[14] realized as a superconducting aluminium waveguide. We cool the waveguide to temperatures of a few tens of millikelvin at which its loss is less than 1 dB per km (refs. 14,45) and its thermal occupation is negligible.

To successfully operate this system, we minimized the heat load at each temperature stage using high-reflectance materials combined with superinsulation for radiation shielding. We designed vertical support structures between the individual shielding stages to minimize thermal conductivity while providing mechanical stability. The system withstands thermal contractions by leveraging flexible thermal connections formed by braids and mobile mechanical supports. We maximize heat flow along the link modules by using high-conductivity copper and minimizing thermal contact resistances between adjacent link elements. At the midpoint between sites A and B, a pulse tube cooler provides an additional heat sink for the thermal radiation incident on the 50 and 4 K radiation shields. At 30 m in length and with a total mass exceeding 1.3 tons of radiation shields cooled to below 80 K, roughly 90 kg of which are cooled to below 50 mK, this constitutes a large-scale cryogenic system operating at millikelvin temperatures[46]; see Supplementary Information section III for details.

At each of the sites A and B, we operate a transmon-style qubit whose state and transition frequency is controlled on nanosecond time scales using amplitude and phase-modulated microwave pulses and magnetic-flux bias pulses. We read out the state of each qubit using a resonator combined with a Purcell filter[12–14]. For the entanglement protocol, we make use of a photon-transfer resonator, also combined with a Purcell filter, which we couple using a coaxial line to the aluminium waveguide connecting the two sites[12,14]. Both qubits and their support circuitry are fabricated on two nominally identical chips (Supplementary Information section IV).

## Bell test

In each individual trial of the Bell test experiment, we deterministically generate a Bell state $|\psi^+\rangle = \frac{1}{\sqrt{2}}(|ge\rangle + |eg\rangle)$ between the stationary transmon qubits at sites A and B using direct photon exchange[12,47] (Methods). Performing quantum state tomography of the states of the qubits at sites A and B, we experimentally achieve a Bell state fidelity of $\mathcal{F}_s^{|\psi^+\rangle} = 80.4\%$, corresponding to a concurrence[38] of 0.765 (Fig. 2b) when correcting for readout errors. The experiments were performed with two independent phase synchronized set-ups separated by 30 m (Supplementary Information section V). Without readout error correction we find $\mathcal{F}_s^{|\psi^+\rangle} = 78.9\%$ and $\mathcal{C} = 0.689$. The concurrence of the entangled state is sufficiently high for violating Bell's inequality (Fig. 2a)

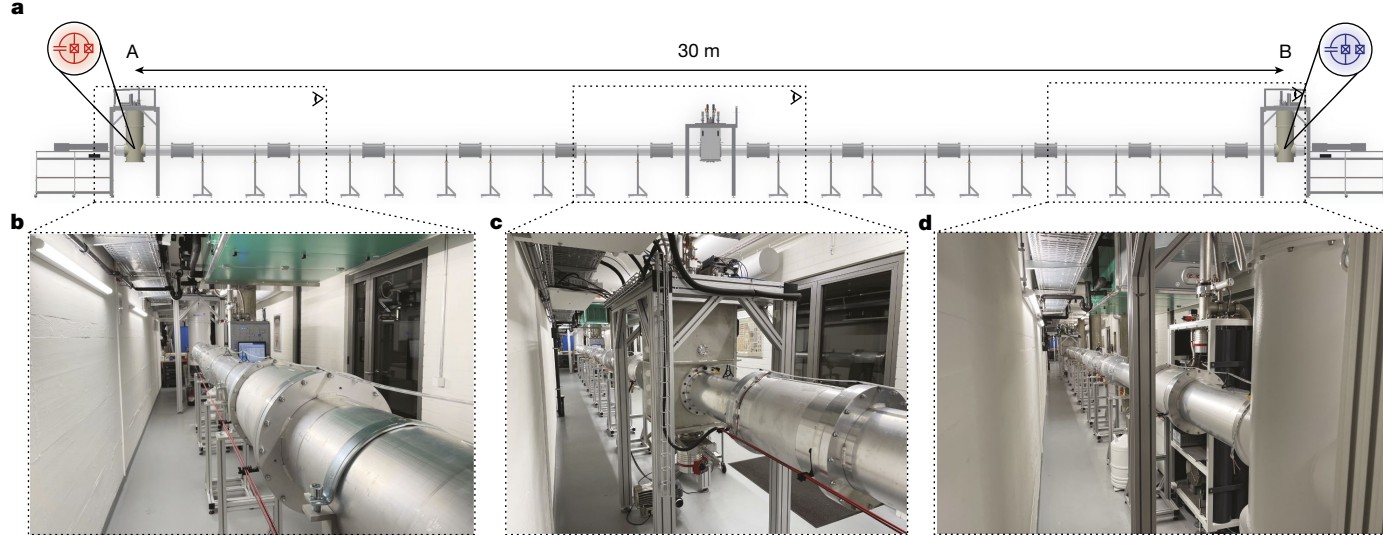

**a**

A      30 m      B

**b**       **c**       **d**

**Fig. 3 | Cryogenic microwave quantum link. a**, Computer-aided design (CAD) model. **b–d**, Photographs of the 30-m-long cryogenic set-up. Dilution refrigerators at each end host the quantum devices that are connected through a waveguide cooled to below 50 mK over the full distance. A central pulse tube cooler provides additional cooling power to the two outermost radiation shields. The photographs are taken at the position of the corresponding eye pictograms shown in **a**. A, (**b**), centre (**c**) and B (**d**).

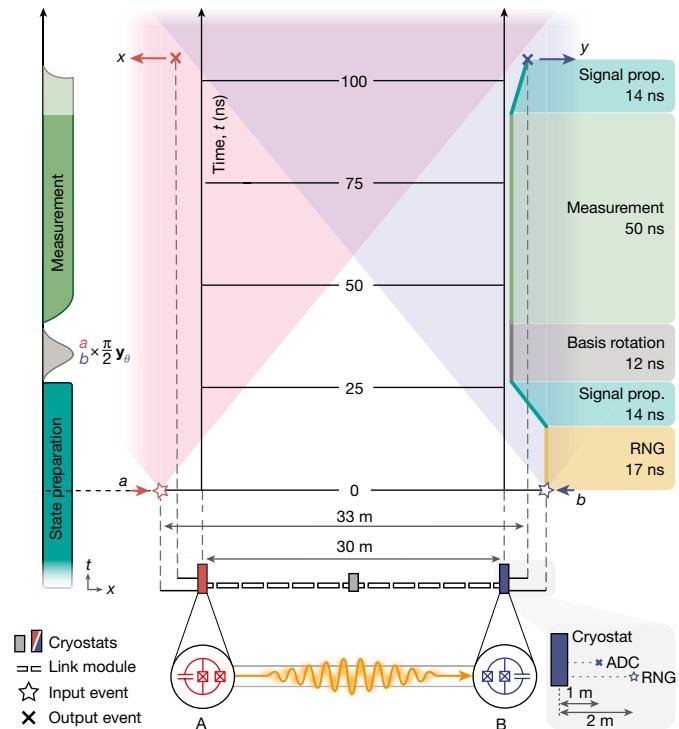

**Fig. 4 | Space–time diagram of the experiment.** The left, vertical time axis schematically shows the microwave pulses applied to the qubits locally at each node. The right axis indicates the duration of the individual Bell test protocol segments: RNG, signal propagation (prop.), qubit basis rotation and measurement. The space–time location of the start and stop events of a Bell test trial are marked with stars and crosses, respectively. The red and blue regions indicate the future light cones of the start events. The inset on the bottom right indicates the approximate spatial location of the start and stop events in the RNG and ADC, relative to the vertical centre axis of each cryostat.

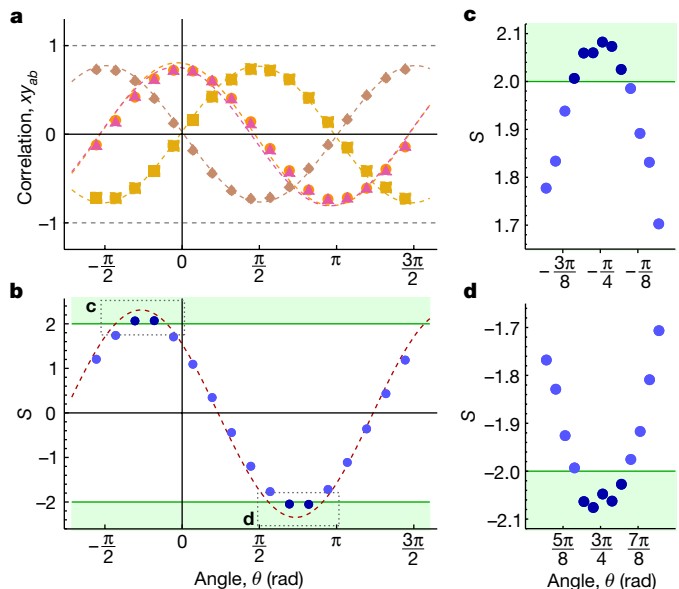

**Fig. 5 | Bell inequality violation versus measurement basis offset angle.**
**a**, Quantum correlations $\langle xy \rangle_{(a,b)}$ of individual Bell tests versus offset angle $\theta$. The 17 data points are results of individual Bell tests with $n_{max}/17 = 61{,}680$ trials each, incremented by $\theta = \pi/8$. The dashed curves are calculated using a master equation simulation. **b**, Corresponding $S$ values calculated from the data shown in **a**. Points are experimental data, and the dashed red line is extracted from a master equation simulation. Error bars are roughly on the order of the marker size, see text for details. **c**, Measured $S$ values of 13 individual Bell tests, with $n_{max}/13 = 80{,}659$ trials each, and offset angles around the expected optimum value $\theta_{S_{max}}$ incremented by $\theta = \pi/32$. **d**, Same as in **c** but for the expected optimum value $\theta_{S_{min}}$. The green lines in **b**–**d** mark the threshold value $|S| = 2$, and all points in the green shaded region correspond to Bell tests that violate the CHSH inequality.

and is on par with previous experiments using the same approach for entangling qubits in a single cryostat[12] and in two cryostats connected across a distance of 5 m (ref. 14). The infidelity is dominated by photon loss induced by a circulator used to extract photons from the waveguide for characterization of the entanglement protocol in all three experiments[12,14] (Supplementary Information section VI). As outlined in the Methods section, we rotate each qubit state along the $y$ axis of the Bloch sphere to maximize the Bell violation. After entangling the distant qubits at sites A and B, we are set to perform the timing-critical part of the loophole-free Bell test as indicated in the space–time diagram in Fig. 4.

To generate the input bits a and b for the measurement basis choice we use a RNG at each node[48]. We consider the start event of each trial of a Bell test as being marked in space and time by the earlier of the two events corresponding to the creation of a random number in each RNG. At node A (B), a random number is generated at the location indicated by a red (blue) star in the space–time diagram in Fig. 4 at a distance of about 2 m from the corresponding qubit housed in its dilution refrigerator. The random number a (b) becomes available as a voltage pulse at the output of the RNG $17.10 \pm 0.14$ ns after this event (yellow section in Fig. 4). This pulse controls a microwave switch that conditionally passes a microwave basis-rotation pulse to the qubit at A (B). We provide additional information on the random basis choice in the Methods section and in Supplementary Information section II.

To achieve a signal propagation delay of the basis choice pulses applied to each qubit of only 14 ns (first turquoise section in Fig. 4), we pass microwave signals roughly along the line of sight connecting the qubits at A and B from the room-temperature switch through a side-access port into the cryogenic system (Supplementary Information section III).

The random basis selection pulse (grey) applied to the qubit has a duration of 12 ns.

After the microwave pulse has fully rotated both qubit states into the randomly chosen basis, we read out the qubits at A and B by applying a microwave tone to their dedicated readout resonators. We detect the amplitude and phase of the readout pulse after several stages of amplification (green section in Fig. 4), record it with a digitizer (analogue-to-digital converter, ADC) and postprocess the data with a field programmable gate array (FPGA)[13]. We achieve single-shot readout fidelities of $\mathcal{F}_r^A = 99.05\%$ and $\mathcal{F}_r^B = 97.60\%$ in only 50 ns integration time (Supplementary Information section VII).

As done with the random basis choice signals, we route the readout signals through the side ports of the dilution refrigerators at sites A and B. In this way, we minimize to 14 ns the propagation delay towards the ADC and FPGA, located at about 1 m physical distance from the qubits, indicated by a cross in Fig. 4. We consider the measurement result to be fixed at time $t_x$, the moment when the last part of the measurement signal arrives at the input of the ADC to be digitized, see Supplementary Information section VIII B for a discussion of this choice.

In each Bell test experiment, we recorded the basis choices $(a, b)$ and the corresponding readout result $(x, y)$ for all $n$ trials. On the basis of these values, we calculated the averages $\langle xy \rangle_{(a,b)}$ for all four combinations of measurement basis choices taking into account all $n$ trials, thus closing the fair-sampling loophole[40,41].

In each of four consecutive experiments, we ran $n_{max} = 2^{20} = 1{,}048{,}576$ individual trials of a Bell test for a total time of about 20 minutes. In the first experiment, we swept the angle $\theta$ between the two randomly chosen measurement bases (Fig. 1) across a full period. Plotting $\langle xy \rangle_{(a,b)}$ for all four input bit combinations versus $\theta$ we observe the expected

sinusoidal oscillations[4], offset from each other by π/2 (Fig. 5a). Ideally, $\langle xy \rangle_{(a,b)}$ oscillates between +1 and −1. The observed reduction in contrast is due to the finite concurrence of the initial entangled state and read-out errors. We note that we have calibrated out an experimental phase offset of $\theta_0 = 160.0°$ between the two sites (Supplementary Information section IX). We find good agreement between the experimental data and a master equation simulation (Fig. 5a,b and Supplementary Information section VI).

On the basis of these data, we then calculate the $S$ value as a function of $\theta$ and observe its sinusoidal oscillation with a maximum and a minimum of $S$ occurring at $\theta_{S_{max}} = -\pi/4$ and $\theta_{S_{min}} = \pi - \pi/4$ (Fig. 5b), offset by π as expected. Here, we evaluate roughly 60,000 trials for each angle. We find that $|S|$ exceeds two at both values $\theta_{S_{max/min}}$, violating the Bell inequality.

Near $\theta_{S_{max/min}}$ we perform a set of measurements with step size $\theta = \pi/32$ and determine the $S$ value from roughly 80,000 trials at each value of $\theta$. We observe that several data sets clearly violate the Bell inequality for angles around $\theta_{S_{max/min}}$ (Fig. 5c,d). From the data set taken at the offset angle $\theta_{S_{max}}$ we find a maximum violation of $S = 2.082 \pm 0.012 > 2$.

In a final experiment performed at the single angle of $\theta_{S_{max}}$, we acquire Bell test data for $n_{max}$ trials, yielding $S = 2.0747 \pm 0.0033$, which exceeds two by more than 22 standard deviations. In this experiment with superconducting circuits, we reject the null hypothesis corresponding to Bell's inequality being satisfied with a $P$ value smaller than $10^{-108}$ (Methods), which is small in comparison to $P$ values reported for Bell tests closing all major loopholes in the literature (Supplementary Information section VIII A). The statistical method used here is robust to memory effects (Supplementary Information section X).

Finally, we verify that the locality loophole is closed by measuring the physical distance $d$ separating the two pairs of points in space marked by stars (Fig. 4) defining the start of the Bell test trial at $t = t_\star = 0$ from the points in space marked by stars defining the end of the trial at time $t_\times$. Using the methods described in the Supplementary Information section XI, we find the shorter of these two distances to be $d = 32.824$ m $\pm 4.6$ mm yielding a time budget of $t_d = 109.489 \pm 0.015$ ns for the Bell test to close the locality loophole. Using independent measurements, we determine the total duration of the Bell test trial to be $t_\times - t_\star = 107.40 \pm 0.26$ ns $< t_d$ (Supplementary Information section XI), therefore closing the locality loophole with a margin of roughly eight standard deviations. These timing margins are similar to those achieved in loophole-free Bell tests reported in the literature (Supplementary Information section VIII B).

To formulate our conclusion, we assume that we can precisely determine the space–time description of the events at hand and that the RNG produced independent, free random bits. Ultimately, such assumptions constraining the conclusion cannot be fully avoided, even in principle[18]. Under these assumptions, we find that our observation of the violation of Bell's inequality with superconducting circuits is incompatible with an explanation satisfying the principle of local causality.

## Performance and outlook

Previous loophole-free Bell tests using polarization-encoded optical photons as qubits, typically violated Bell's inequality with a lower margin[7–9] than our experiment ($S = 2.0747$) whereas atomic and solid-state systems[6,10] realized higher violations. By reducing the loss in the channel connecting the two qubits in our set-up and thus increasing the Bell state fidelity, we expect Bell violations with $S > 2.4$ to be achievable in future experiments while closing all major loopholes. We plan to reach this target by omitting the circulator from the waveguide, as in ref. 37, and using both a low-loss printed circuit board and superconducting microwave cables connecting the sample mount to the waveguide. With these measures, we estimate to reduce the photon loss by up to a factor of four to about 5%. Alternatively, a heralded entangling method, avoiding loss but effectively reducing the repetition rate, can be implemented for the same purpose[12,49]. Such improvements may enable protocols requiring larger Bell violations, such as device-independent quantum key distribution[50], to be executed between superconducting quantum processors connected in a network.

Because the experiment presented here operates at a repetition rate of 12.5 kHz, which is larger than those of loophole-free Bell tests with atomic and other solid-state systems[6,10], we achieve highly statistically significant Bell inequality violations in only a few minutes. This is similar to experiments performed with polarization-encoded optical photons that have even higher repetition rates[7–9]. Further improvements in the repetition rate of our experiment seem feasible, up to the inverse of the duration of the pulse sequence used. In Supplementary Information section VIII we compare in detail the performance metrics of published Bell tests that also used a minimal set of assumptions.

To implement device-independent quantum information processing protocols, it is desirable to simultaneously achieve high Bell violations and high repetition rates. The set-up demonstrated in our experiments provides an interesting combination of those metrics allowing us to visualize implementing a variety of device-independent quantum information processing protocols[26–30] with superconducting circuits, a promising candidate for large-scale quantum computing systems[11,31].

In addition, our experiment demonstrates that quantum information can be transmitted between superconducting circuits housed in cryogenic systems separated by tens of metres, going beyond our previous work on a metre-scale system[14]. Interconnected cryogenic systems may indicate a pathway towards realizing larger scale quantum computing systems using quantum microwave local area networks[51], for example, within a quantum computing centre. The set-up also enables the exploration of non-local quantum physics with degrees of freedom that couple to microwave photons such as mechanical resonators or spins.

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

# Methods

## Superconducting qubits

For completeness, we discuss a few key features of superconducting quantum electronic circuits here. Superconducting qubits are anharmonic quantum oscillators with circuit parameters chosen to realize resonance frequencies in the gigahertz frequency range (ref. 52 and references therein). The non-linearity of the circuit is provided by an ideally lossless Josephson element[53], which, in our experiments, is realized as a pair of Josephson tunnel junctions arranged in a superconducting quantum interference device loop shunted by a large capacitor jointly forming a transmon qubit[54]. The effective Hamiltonian of the circuit is governed by a cosinusoidal potential that hosts a set of bound states, the lowest two of that form the computational basis states of the qubit (labelled $|g\rangle$ and $|e\rangle$). The second excited state (labelled $|f\rangle$) can be used, for example, as an auxiliary state to realize two-qubit gates[55], or, as in this paper, for emitting photons with a controlled temporal mode profile[12,47] when coupling the qutrit strongly to a superconducting resonator (for example, ref. 56 and references therein). Here, as well as elsewhere in this publication, we refer to the quantum bit as a qutrit when we refer to its lowest three energy eigenstates. To minimize thermal excitation of the qubit and also minimize losses in the superconducting materials used to realize the qubits and the 30-m-long waveguide, we operate the devices at temperatures around 15 mK (ref. 25).

## Generation of the remote, two-qubit entangled state

We generate the Bell state $|\psi^+\rangle = (|ge\rangle + |eg\rangle)/\sqrt{2}$ between the two remote qubits A and B using a deterministic scheme based on the exchange of a single photon[57], as demonstrated with superconducting circuits, for example, in ref. 12. In this protocol, qubit A is first entangled with a propagating, ideally time-reversal-symmetric photon in a driven coherent emission process. The propagating photon is then deterministically absorbed in a time-reversed process at qubit B creating the desired entangled state.

The pulse sequence used in the process is about 400 ns long and ends at roughly $t = 16$ ns after the process initiating the random basis choice starts, see Supplementary Information section VI for details. We create the entangled state as an input resource to the presented Bell test experiment. We consider the entanglement process itself to be independent of the timing constraints of the loophole-free Bell test (Methods and Supplementary Information sections II and XI).

We characterize the created Bell state using quantum state tomography (Fig. 2b) for which we rotate the density matrix in postprocessing by an angle $\theta_0$ around the $z$ axis to maximize the Bell state fidelity. $\theta_0$ is the experimental offset angle between the two set-ups A and B (Supplementary Information section IX). We also perform a master equation simulation of the Bell state generation protocol and find good agreement with the experimental data (red wireframes in Fig. 2b), characterized by the small trace distance $\sqrt{\mathrm{Tr}(|\rho - \rho_{\mathrm{sim}}|^2)} = 0.077$.

## Optimizing the measurement basis

Photon loss and qubit decay are the dominant mechanisms reducing the fidelity of the experimentally created Bell state. These processes create an asymmetry between the qubit excited and ground states participating in the Bell test. To reduce this detrimental effect and maximize the $S$ value, we perform the Bell test by choosing measurement bases in the $xy$ plane of the Bloch sphere, where each basis is affected equally by photon loss and qubit decay. To do so, we apply a $(\pi/2)_x$ basis-rotation pulse to qubit A, and a $(\pi/2)_{x+\theta}$ pulse to qubit B in each experimental trial just after the $|\psi^+\rangle$ Bell state preparation and before applying the pulse implementing the random measurement basis choice. Here, $\theta$ denotes the angle between the two randomly chosen measurement bases, as introduced in the main text. Effectively, this pulse sequence generates the Bell state $|\phi^+\rangle = (|gg\rangle + |ee\rangle)/\sqrt{2}$.

## Random basis selection and measurement independence

As in previous loophole-free Bell tests[6–8,10], we use well-characterized, fast physical RNGs[48] to support the measurement independence assumption. The basis choice bits a and b are generated by a dedicated RNG at each node. Each RNG contains eight quantum entropy sources, each composed of a laser to produce phase-randomized pulses, an interferometer, fast linear detection, one-bit digitization and a parity calculator. We assume that the extracted random bits are independent from all previous events. Supplementary Information section II describes support for this assumption and in Supplementary Information section X we perform a statistical analysis.

The RNG output bit is encoded as a voltage, and controls a single-pole single-throw microwave switch, which (with input high) blocks or (with input low) passes a microwave $(\pi/2)_y$ pulse to induce a basis change of the corresponding qubit. We discuss the basis selection further in Supplementary Information section II.

Because every choice of measurement leads to a recorded experimental trial in our experiment, as in refs. 7,8, the RNG properties offer direct support for the validity of the measurement independence condition. This differs from the situation in event-ready Bell tests[6,10], in which a heralding event—the result of a joint photon measurement at a middle station—determines whether or not a given experimental trial is recorded for analysis. Such selection opens the possibility for the heralding event to select trials as a function of the measurement choices, if these are not space-like separated from the joint measurement. In this situation, the measurement independence condition can be violated even in presence of perfect RNGs. For this reason, in event-ready Bell tests it is important that the heralding event is space-like separated from the events marking the creation of the random-input bits. Because we use a deterministic entanglement generation protocol that is not dependent on the outcome of any measurement, such considerations related to a heralding event do not play a role in our experiment.

Note that when focusing on specific families of local hidden variable models, measurement independence can sometimes be supported by space-like separation alone. This is the case for the model introduced in Scheidl et al.[58], and relevant for Bell tests with entangled-photon pairs, where the hidden variable $\lambda$ is assumed to be created at the photon pair generation event independently of any past event. With that assumption, space-like separation between the pair generation and the measurement choice events guarantees measurement independence. This space-like separation is achieved also in refs. 7,8. Assuming $\lambda$ is produced along with the photon pairs, but relaxing the assumption that $\lambda$ is independent of past events, a photonic Bell test with the same space-like separation between the pair generation and the settings choice events can exclude local causal models in which the photon pairs influence the measurement choices, but not local causal models in which earlier events influence both the photons and the basis choices. Because such past influences could be arbitrarily far in the past, space–time conditions cannot be used to fully support the assumption of measurement independence beyond the model of Scheidl et al.[58], even in photonic Bell tests. For this reason, we do not attempt to create a space-like separation between the entanglement generation and the basis choices, but rather support the assumption of measurement independence by using well-characterized RNGs[48], as has been done in previous Bell tests since the pioneering work of Weihs et al.[20].

## The $P$ value as a statistical metric

Early Bell test experiments typically used the standard deviation as a metric to discuss the statistical significance of an observed Bell inequality violation. This approach, however, comes with two limitations. The first is that when using the notion of standard deviation, we implicitly assume that the underlying measurement data are Gaussian distributed. This assumption is only justified in the limit of infinitely many trials, but in experiments a finite number of trials are executed.

A statistical analysis of the Bell test based on standard deviations may therefore overestimate the statistical significance of the result[5,59]. The second limitation is that the notion of standard deviation relies on the assumption that the result of the $k$th trial is independent of the basis choices and measurement results of the previous $k-1$ trials, which opens up the memory loophole[18]. These two limitations can be addressed by the statistical analysis of the result through the calculation of a $P$ value according to a method that does not rely on any of the aforementioned assumptions. Therefore, the calculation of $P$ values in the context of Bell tests is now an established practice[6–10]. In this context, the $P$ value is a metric of the probability with which data as extreme as those observed could have been produced by a local causal model (see Supplementary Information section X for details).

## Data availability

All data are available from the corresponding authors upon reasonable request.

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

**Acknowledgements** We thank M. Frey, N. Kohli, R. Schlatter, A. Fauquex, R. Keller, B. Dönmez, M. Hinderling, S. Wili, F. Marxer, A. Schwarzer and J.-A. Agner for technical support on setting up the laboratory, and designing, building and testing the cryogenic system. We thank J. Herrmann, L. Raabe, N. Mostaan, E. Portolés, M. Ruckriegel and J. Heinsoo for their contributions to software and electronics. We thank A. Aspect, G. Blatter, N. Gisin, R. Hanson, A. Imamoglu, R. Renner and R. Wolf for commenting on an early version of the manuscript. The work at ETH Zurich was funded by the European Research Council through the 'Superconducting Quantum Networks' (SuperQuNet) project, by the European Union's Horizon 2020 FET-Open project SuperQuLAN (grant no. 899354), by the National Centre of Competence in Research 'Quantum Science and Technology' (NCCR QSIT), a research instrument of the Swiss National Science Foundation, and by ETH Zurich. B.R. and A.B. acknowledge support from the Natural Sciences and Engineering Research Council of Canada, the Canada First Research Excellence Fund and from the Vanier Canada Graduate Scholarships. J.-D.B. and N.S. acknowledge support by the Institut de Physique Théorique, Commissariat à l'Energie Atomique et aux Energies Alternatives, by the European High-Performance Computing Joint Undertaking under grant agreement no. 101018180 and project name HPCQS and by a French national quantum initiative managed by Agence Nationale de la Recherche in the framework of France 2030 with the references ANR-22-PETQ-0007, project name EPIQ and ANR-22-PETQ-0009, project name DIQKD. M.W.M. acknowledges support by NextGenerationEU (grant no. PRTR-C17.I1) and by projects SAPONARIA (grant no. PID2021-123813NB-I00) and MARICHAS (grant no. PID2021-126059OA-I00), by 'Severo Ochoa' Center of Excellence CEX2019-000910-S, Generalitat de Catalunya through the CERCA program, AGAUR grant no. 2021-SGR-01453, by Fundació Privada Cellex and by Fundació Mir-Puig.

**Author contributions** S.S., J.S., A.K. and P.M. planned and performed the experiment and analysed the data. S.S., P.M., T.W. and J.S. designed and tested the quantum devices. J.S., J.L., S.S., P.M., P.K. and A.W. devised and tested the cryogenic set-up. J.-C.B., M.G., G.J.N., T.W. and A.R. fabricated the devices. P.M. and S.S. integrated the measurement basis choice scheme. A.K., A.C., S.S., J.S. and P.M. developed and implemented the set-up synchronization scheme and the verification procedure for closing the locality loophole. S.S., A.C., K.R., P.M. and A.A. developed the experiment control and FPGA-based data analysis code. B.R., S.S. and A.B. ran the master equation simulations. F.M., J.M., W.A., M.W.M., C.A. and P.M. developed and tested the RNGs. J.-D.B. and N.S. assisted with the statistical analysis and the manuscript. S.S., J.S. and P.M. provided the figures for the manuscript. S.S. and A.W. wrote the manuscript with major contributions from A.K., J.S., J.-D.B., N.S. and M.W.M. and input from all authors. A.W. supervised the project.

**Funding** Open access funding provided by Swiss Federal Institute of Technology Zurich.

**Competing interests** The authors declare no competing interests.

**Additional information**
**Correspondence and requests for materials** should be addressed to Simon Storz or Andreas Wallraff.
