## [Peer Review File · Nature]

Manuscript Title: Loophole-free Bell Inequality Violation with Superconducting Circuits

Reviewer Comments & Author Rebuttals

Reviewer Reports on the Initial Version:

Referees' comments:

Referee #1 (Remarks to the Author):

Summary. The paper performs a loophole-free Bell experiment with entangled qubits realized by superconducting circuits separated by 30 meters.

Originality and significance. Loophole-free Bell experiments are exceptionally challenging technically, requiring long-distance high-quality entanglement of qubits with ultra-fast measuring schemes that can toggle between different measurement angles in timeframes measured in nanoseconds. Accordingly, the first such experiments were not performed until 2015 and no more than a handful of experimental groups have achieved the loophole-free benchmark to date. Loophole-free Bell experiments are important because only they can definitively lay claim to exhibit nonlocality as a physical phenomenon independent of any assumptions about the correctness of quantum mechanics and/or the quantum modeling of the experiment, and they are required for the execution of the most secure device-independent quantum communication protocols. The 2022 Nobel Prize was awarded largely for, in essence, early experimental work towards the goal of loophole-free Bell tests.

The current experiment is notable in being the first loophole free Bell experiment employing superconducting circuits. (Previous experiments had used photons, NV centers, and neutral atoms). While demonstrating a loophole-free Bell experiment in any new platform is a major scientific achievement, this experiment can lay special claim to additional significance of having done this with the same platform as used for quantum supremacy experiments reported in Ref. [10] and [Phys. Rev. Lett. 127:180501 (2021)]. Thus the current experiment shows that 1) a quantum advantage outperforming classical computers (per the above references, modulo any potential critiques/re-assessments for which I'm not up-to-date) can be complemented with 2) a quantum advantage in exhibiting nonlocal correlations -- arguably THE two main superclassical quantum science capabilities of the second quantum revolution -- all in one system of superconducting qubits. I think this is fascinating, not just because of the potential for applications towards implementing integrated quantum information protocols, but on a more pure science level of stimulating thought on the nature of quantum advantages and the not-completely-understood link between the two types (computation and nonlocality). Indeed I believe the manuscript could have done a bit of a better job alluding to this deeper exciting significance along the lines of what I am loosely trying to get at here.

In light of this, I recommend this manuscript for publication in Nature, provided the authors can satisfactorily address some mostly minor clarifications that I ask about below.

Data & methodology. The overall scientific quality is high. The authors rightly recognize the importance of precise timing. I do have a few requests for clarification regarding this.

First, the start event: The start event is characterized as the time the "laser goes above threshold" (line 808); to confirm in layman's terms, is this as the time when (or at least no later than when) the laser starts emitting the photons that will then be measured to obtain the randomness for settings? And since this time can't be measured directly during runtime, how exactly do you obtain the 17.10+- .14 ns figure (807) for the delay until the observable reference event t_0 ? The discussion (812-815) suggests a characterization of the internals of the device, which may be OK, but a little more information/clarity would be helpful here. And can the authors address how this choice of start event compares to the choice of start event for the other LHF tests, just as they did a comparison for the choice of end event - was the start event the same as for these other experiments (which used similar/same RNGs)?

Second, concerning the establishment of a common laboratory reference frame (898-899). This is of key importance; when determining the Bell time budget at Alice, you need to measure the difference of time coordinates between 2 events, the latter of which is defined as the time a specific spatial location at Alice enters the forward light cone of an event that occurs 25 meters away (at Bob). Thus, to measure the duration of this window, you need a very precise synchronization of clocks at Alice and Bob to sub-nanosecond precision. The authors say that this is done by "sending a square pulse from a central trigger device" (900) to the other devices, then enter into a long terminology-filled discussion that I don't have the technical expertise to follow. I suspect this took more effort than (just) making sure "output signals arrive simultaneously through equally long [12.5 m?] cables" (right?). Overall I am willing to defer to the experimental expertise of the team, if they can re-affirm in the reply that they are capable of synchronizing clocks separated by 25m to within fractions of a nanosecond with super-high confidence, and that when measuring durations of various steps (fig 9) that they are carefully accounting for how long various signals take to travel (1 m \geq 3 ns!) from the thing whose duration they are measuring to whatever nearest clock that is keeping time (I think the three oscilloscopes in 1142?). I will state that the descriptions of measurement procedures that I can better understand and follow (for instance Appendix E.2 measuring distance from Alice to Bob) meticulously took into account everything appropriately and made some sound conservative choices.

Third, there are aspects of photon generation/detection that surprised me and I want to ask about. This request is just a sanity check on my own part and does not require changes in the paper if the authors feel these aspects would be unsurprising to a specialist, which I would suspect is the case. So: I understood from the manuscript that prior to every trial, to create entanglement, a single microwave photon is emitted from one party to the other along the 25m long waveguide (490, 1387). This is a little surprising to me because I thought it was hard to create single photon on demand sources: I'm familiar with low-power (optical) lasers in the LHF bell tests Ref [6] and [7] where if the low-power laser was emitting on average one photon during a time unit you would have a Poisson distribution leading to a large probability of zero-, two-, three- photon events with the higher-number events fairly frequent (maybe 30% probability) and especially detrimental to the Bell test. If I understand (1461-1463) correctly, higher-number events are not unknown to your setup but these must be happening at much lower probability. So is your setup just fundamentally different

where on-demand single-photon generation is not difficult? On a related note, it is my understanding that single photon detection is hard, but it seems like your apparatus for creating remote entanglement effectively detects the incoming photon at B with very high efficiency (1398), or more precisely could be converted into a very efficient photon detector with perhaps different choice of measurement if that was the goal instead of creating entanglement. Can you comment on this as well? It seems to me like possible answers are a) microwave is different than optical frequency, 2) waveguides (or ultra-cold waveguides) are different than optical fiber and/or free space 3) lasers are different than your photon source 4) actually, it's not just one photon traveling from Alice to Bob prior to each trial to create the entanglement, it's a whole bunch.

Appropriate use of statistics and treatment of uncertainties. The statistical method detailed in Appendix L is appropriate and the computed p-value 10^{-108} looks consistent with the appropriate figure (L1), which is the (extremely small) probability of a Binomial random variable with $p(\text{success})=0.75$ exceeding 796228 successes in 2^{20} trials (a 0.75934 success rate). Putting error bars on the CHSH violation is fine when considering this as a measurement of a physical parameter but statements using this uncertainty bar to quantify violating Bell's inequality ("by 22 standard deviations" in abstract, see also line 339) suggest this as a stand-in for a p-value for violating locality, which is incorrect. While mostly harmless, I recommend removing these statements. On a related note, the use of standard deviations in lines 1715-1723 as a sort of metric to measure the absolute size of Bell violations doesn't make sense - what matters in this context is comparing the the absolute sizes (eg, 2.6 is better than 2.4) while acknowledging the precision of each figure; but 2.4 with better precision (possibly just meaning more statistics were taken) isn't suddenly better than 2.6 because it is more standard deviations above 2.

Conclusions. Overall, exceptional. I do have some concerns about how the "principle of locality" is presented, but these can be easily fixed with a better discussion. Specifically: the statement that the principle of locality "derives from the observation that no causal influence can propagate faster than the speed of light" (38-39, see related at 55-56) may be misinterpreted to suggest that Bell violations imply causal influences faster than light. Indeed the "principle of locality" is not term with a standard well-accepted definition in the literature, and the proffered definition in Appendix A of "allowing events to be influenced by an arbitrary action in their past light-cone" (624) is faulty (surely QM "allows" this). A better definition, perhaps rooted in local hidden variables terminology, should be formulated. Note, I agree with the authors that "realism" is a term that is perfectly fine to avoid. Other things: The locality loophole is opened when relevant events aren't outside light cones, not the statement (604-605) which would incorrectly suggest that QM exploits the locality loophole to violate Bell inequalities (note A2 is violated by QM while A3 is obeyed by QM). Similarly, lines 609-615 is wrong; space like separation of the events doesn't "guarantee the validity of A2"; see again QM. Finally, I believe that one can claim that conclusions not resting on discussions of realism are clearer, more precise, or better posed (etc.) but I would hesitate to say "stronger" (665) because the fuzziness of the realism concept is the problem, not that there is some clearly defined mathematical definition of realism (something like A1 and A2) that is in fact superfluous.

Suggested improvements. In addition to what I have discussed above, I have the following small edits: (91-92) "making bit strings more random, a task impossible by classical means" doesn't

sound right, maybe reword or just stop sentence at "randomness amplification" if a better description this concept about the "bit strings" (S-V sources) gets too digressive; suggest writing either $t_{\text{star}}=0$ at (116) or $t - t_{\text{star}} < td=d/c$ in (124); I suggest depicting and labeling "d" (from line 122) in Figure 1; I see $|g\rangle$ and $|e\rangle$ defined as ground and excited in (139) but is $|f\rangle$ defined anywhere; strictly speaking this does not require addressing if obvious to a specialist but I am just curious why it is required for the entire 25m waveguide to be so cold (199-201) when optical fibers can transmit photons without too much loss at room temperature; around (343) I would re-mention that the statistical analysis leading to the p-value of 10^{-108} is robust to memory effects and cite appendix L and the references therein for the methods used to perform said statistical analysis and obtain the figure; it is not quite correct to state that high Bell violations are necessary for device-independent protocols (406-409): this is true for DI-quantum key distribution for which there is a threshold strictly larger than 2, but not true for DI-random number generation where any violation above 2 can be used; (A5) remove last "x" from conditioner; define angled brackets notation in (A7); I appreciate that a sanity check was performed for the equiprobability of settings frequencies (774) and a further sanity check to ensure no signaling (independence of Alice's outcome probabilities from Bob's setting choice and vice versa) would be nice to report as was done with at least some of the other LHF experiments; "copper" is capitalized multiple times in the SM; I can guess at the reason but state why signals are attenuated as they travel into the cold regions (865-866); align labeling of "Node C" in appendix D and "Chronos" in Fig 7; in Appendix E.2 fix the many terminology problems - be clear that Bell distance is in fact a time interval in the laboratory reference frame, the norm notation in (990) is mathematically inappropriate, be clear whether the p-vectors are locations or events because you are measuring the physical distance between them in (1003), and a small thing but is there in fact just one additional neglected spatial direction not two (1023) as you are projecting onto the the 2-D floor not a 1-D line; citation of [50] in (1703) seems to be an error.

References. The reference list is comprehensive and the location of citations is overall good with only minor recommended changes mentioned above.

Clarity and context. Generally very clear, and good context is provided. I have one recommendation, that the authors state what IS a superconducting qubit - this was strangely absent from Ref. [10] as well. Perhaps this is technical and hard to describe, but at a minimum, what is the degree of freedom that can exist at two levels to implement a qubit? Also, there is talk of a qutrit that is somehow coupled to the qubit playing some role in sending and receiving photons to create entanglement and/or read out measurement outcomes, I'm curious to hear a little more about this mechanism as well without having to consult references. A full description of these things is obviously beyond the scope of the present manuscript, hopefully there is room for perhaps a phrase or two in the main manuscript stating what the mechanisms are, and maybe a little more in the SM with indications where the mechanisms are located in Fig. 12.

Referee #2 (Remarks to the Author):

This work reports a loophole-free Bell test with quantum measurements on superconducting qubits. To achieve this goal, the authors entangle a pair of superconducting qubits separated by a physical distance of 30 meters. This physical distance is much larger than those between two

superconducting qubits achieved in previous works. At the same time, the two remote qubits reported here are evaluated to be strongly entangled such that a significant violation of the CHSH Bell inequality can be observed accordingly. Moreover, the authors achieve rapid, high-fidelity readout of superconducting qubit states using the state-of-the-art readout scheme optimized in their previous work [Ref. 12]. These achievements enable the authors to close the locality and the detection loopholes for the first time in a Bell test using superconducting qubits. In my personal opinion, the loophole-free Bell test reported in this work is a landmark experiment.

The manuscript provides a good description of most aspects of this work. Nevertheless, there are a few statements which I found confusing or unclear:

1) In the main text and Appendix A, the authors specify the two underlying assumptions in the derivation of Bell inequalities --- locality and measurement independence. I cannot fully agree with this statement. The use of local hidden variables to describe a quantum experiment implies that the measurement outcomes under different settings can coexist or be predetermined before the measurements. To my understanding, the coexistence or predetermination is not captured by the notion of locality but by the notion of realism in the literature. In fact, as shown by Fine [Phys. Rev. Lett. 48, 291 (1982)], probabilistic, local hidden variable models and deterministic hidden variable models are equally powerful. Considering the above, I prefer to say that the two assumptions underlying the derivation of Bell inequalities are local realism and measurement independence.

In addition, I am not sure how to interpret the implication relation stated in Eq. (A9). To me, the meanings of locality and space-like separation overlaps.

2) In the Methods section, it is mentioned that "the duration of the pulse sequence (for entanglement creation) is irrelevant for closing the locality loophole as it occurs before the basis choice is initiated ...". However, I didn't catch a statement in the current manuscript on at which time point the entanglement is nominally created.

A related question is whether the creation of entanglement and the determination of basis choices can be space-like separated in the experiment reported here. I understand that it may be difficult to achieve such space-like separation in the current experiment, as neither a heralded method nor a middle station between Alice and Bob is used for entanglement creation/distribution. However, the readers may still be interested in seeing the comparison between the loophole-free Bell test reported here and those reported in previous works.

3) The description of the entanglement-creation scheme in the Methods section is brief. The working of the scheme would be clear if the joint quantum state of Alice's qubit, Bob's qubit and the intermediate photon could be tracked after each step.

4) In Appendix I, terms like transmission fidelity, transfer efficiency, and transfer fidelity are used. Are they referring to the same thing? If yes, it would be better to keep using one of these terms.

5) In Fig. 14, it is mentioned that "the optimal integration weights are shown in dark blue". However, it is not clear to me in what sense the integration weights are optimal. In addition, I wonder whether

the integration weights and the state assignment thresholds are fixed before analyzing the Bell-test data.

6) The definition of a p-value needs some work. A p-value is the probability, if the null hypothesis (for example, local realism) holds, of yielding a statistic as extreme as the observed one. Note that it is not necessarily equal to the probability of yielding the observed statistics according to the null. Given a Bell-test configuration, there are many different local hidden variable models. In this case, the p-value is defined as the worst-case tail probability. In view of the above, Eq. (L1) and the sentences around it need to be clarified. Several variables in Eq. (L1), for example c and p_{win} , also need to be explained.

Side comment: the upper bound on the p-value obtained in Eq. (L6) is comparable to that according to Theorem 2 (and Corollary 8) in arXiv:1709.04078 by Wills et al.

7) In the end of Appendix K, it is mentioned that "from this data (in Table IV) we can calculate the CHSH S-value and the corresponding statistics". However, there seems to be no details behind in the current manuscript.

A related comment lies in the evaluation of experimental evidence against local realism. From the discussion in Appendix M.1, one can see that the Bell tests performed at the early stage use the number of standard deviations, while the recent loophole-free Bell tests summarize their evidence against local realism into p-values. It may be helpful to point out this and briefly explain the underlying reason, as general readers may wonder why different metrics are used. Note that the potential problems with the number of standard deviations have been discussed in previous works, see Ref. [4] and Phys. Rev. A 84, 062118 (2011) by Zhang, Glancy, and Knill.

I also have a couple of additional suggestions and observe several minor problems as follows:

- 1) It could be helpful to testify whether the experimental data presented in Table IV satisfies no-signaling conditions.
- 2) In Fig. 2(a): It might be better to label the readout fidelity and concurrence without correction as these quantities directly determine the Bell violation observed in the experiment. Is my understanding correct?
- 3) In Abstract: "corresponding to a p-value of 10^{-108} " \rightarrow "corresponding to a p-value no more than 10^{-108} "?
- 4) In Appendix J: "as discussed in Appendix D" \rightarrow "as discussed in Appendix E"? "Fig. J" \rightarrow "Fig. 14"?
- 5) The possible values for the outcomes x, y in Eq. (K1) are not consistent with those in Tables III & IV.
- 6) Besides Refs. [75, 76], another experiment demonstrated device-independent randomness expansion. See the work [Phys. Rev. Lett. 126, 050503 (2021)] by Li et al.
- 7) The work Liu et al. (2018) reported a p-value for rejecting local realism as small as $10^{-204792}$.
Side comment: For the other demonstrations of device-independent randomness generation/expansion mentioned in Fig. 16, the p-values for rejection should be also extremely small, although they were not reported explicitly.

Overall, I consider the loophole-free Bell test reported in this work significant and impressive. I

would therefore be happy to recommend acceptance, provided that the points mentioned above can be addressed or clarified.

Referee #3 (Remarks to the Author):

Nature: Loophole-free Bell Inequality Violation with Superconducting Circuits

Summary and significance: The authors carry out an experiment in which they violate a CHSH Bell Inequality while addressing a combination of major loopholes, including the “locality”, “fair-sampling”, and “memory” loopholes. Their impressive setup consists of two superconducting qubit circuits, each coupled to its own control and readout electronics, and each coupled to opposite ends of a 30-m superconducting waveguide. All superconducting elements are contained within a single, large vacuum cryogenic system. The methods and supplemental materials offer a comprehensive analysis of the experimental design, construction, and data analysis, including a comparison to previous Bell experiments that have been labeled “loophole-free”. Although other setups in the past few years have been shown to close these loopholes simultaneously, this is the first experiment on the superconducting circuit platform to both address the locality loophole and close the fair-sampling loophole, and to my knowledge, it represents the largest physical separation demonstrated to date between two entangled superconducting circuits. Given the foundational significance of Bell-Inequality violations, and the promise that superconducting qubits have for realizing quantum computing technology, it is of great value to the community to see such an experiment on this platform.

The performance of this experiment is noteworthy compared to previous experiments. Photon-based implementations historically show low margins of violation with high data rates, which allow them to reach high statistical certainty of the violation; matter-based implementations historically show higher margins of violation but suffer from low data rates, which can make it difficult or at least time-consuming to reach high statistical certainty. This implementation yields a higher margin of violation than previous optical setups, and a higher data-rate than previous matter-based setups. It includes strong statistical claims generally in line with the strongest previous experiments.

Comments:

Defining Locality

The authors state in the first two sentences of the abstract (lines 18, 19) that, “Superposition, entanglement and non-locality constitute fundamental features of quantum physics” and “quantum physics does not follow the principle of locality”. Einstein’s locality, represented in special relativity theory, states that an object is directly influenced only by its immediate surroundings. Violations of Einstein-locality would enable signaling or faster-than-light communication of messages; this is decidedly not a feature of quantum theory. Quantum theory follows Einstein-locality.

The aspect of quantum theory EPR and Bell focused on in refs 1 and 2 is the situation that, in a bipartite quantum system, the joint outcome probabilities don’t factor. EPR and Bell investigate whether the addition of so-called “hidden variables” could enable each component system to be

described independently. The worldview in which such hidden variables exist and are governed by Einstein-locality has been given many names even within the quantum community, for clarity for the moment, let's call it Bell-locality. Bell showed that no model using (Einstein-)local hidden-variables (LHVs) can reproduce all predictions of quantum mechanics, that is, quantum theory violates Bell-locality.

The distinction between Einstein-locality and Bell-locality is crucial, and is currently missing from this manuscript. There are three aspects of this discussion I'd like to see addressed.

1) Clearer definitions and language around "locality" throughout the main paper's text. Even if the manuscript were targeting only a small community of Bell-locality enthusiasts, it would benefit from clearer verbiage; given that the manuscript is targeting a broader audience, more specific language and definitions are critically needed. Throughout the manuscript, the authors use the general term "locality" to mean the more nuanced "Bell-locality" and "local models" to refer to LHV models. In fact, Einstein's locality predates Bell's, and a Wikipedia search for "principle of locality" describes Einstein-locality; an uninitiated reader could reasonably be confused or misled by the jargon, interpreting the manuscript to claim that quantum theory violates Einstein-locality—which is false. The manuscript would benefit from expressing clearly that the thing we actually test is a set of LHV models.

2) Cleaner statements about "realism" in the appendix. The end of Appendix A (line 657) includes a specific assertion that "no notion of realism appeared in our discussion so far, neither in the derivation of a Bell inequality, nor in the definition of the model. This shows that this concept is not necessary to interpret the result of a Bell test, as already noticed by Bell and others" In fact, the derivation of Bell's inequality in the original and CHSH forms include assumptions of Einstein-locality and determinism—the second component is left out of the discussion in Appendix A, but the omission from this manuscript does not remove it from relevance. In other works, some authors use the term "local realism" as a synonym for "Bell-locality"—meaning any theory of LHVs—as a way to differentiate from Einstein-locality. In fact, however we call it, a violation of Bell's inequality represents a violation of all covered LHV models, not the broader set of Einstein-local models. I'd encourage the authors to reconsider this paragraph, both in content and tone. At a minimum, the authors should please clarify that, even with their chosen notation/definition for "locality", a violation of Bell's inequality does not imply a violation of special relativity.

3) Reference 37 (line 2015) needs some attention. It references the entire "Speakable and Unspeakable in Quantum Mechanics" book, which is a compendium of Bell's papers on the topic. I recommend two adjustments:

- a) The book is a compendium of 24 papers. The citation should include a more specific reference—if not the page number then at least the paper title.
- b) Aspect, who wrote the forward, is listed as a co-author. However, the book is entirely Bell's work. Unless the reference is intended to cite something in the forward, Bell should be the author.

Inconsistent definition of "measurement independence"

The term "measurement" is an ambiguous word, especially in quantum contexts, as it may not distinguish between measurement inputs, outputs, time, processes, parties, etc. Even within this

manuscript, the term “measurement independence” is used inconsistently. I invite the authors to both remove the inconsistency and possibly replace the term “measurement independence” with something more specific.

Line 57: “measurement independence, the idea that the choice of the two possible measurements is statistically independent from the qubits and the measurement devices”

What does it mean for the choice to be independent from the qubits and measurement devices? Independent from possible LHVs located in the qubits? From measurement outcomes? Something else?

In this case, between lines 53 and 60 I think the authors are trying to capture the assumptions of

- 1) independence between each setting choice and the distant measurement outcome
- 2) independence between each setting choice and any hypothetical hidden variable and the text could be refined to clarify this.

Line 573: “measurement independence condition $P(\lambda|ab) = P(\lambda)$ ”

Here, the authors explicitly define the condition as independence between each setting choice and any hypothetical hidden variable (#2 above).

Missing/incomplete discussion of setting-LHV independence.

Two kinds of setting independence are assumed in any physical implementation leveraging Bell’s inequality to test a local-hidden-variable (LHV) worldview.

1) setting-outcome independence: Independence between Alice’s (Bob’s) measurement setting choice and Bob’s (Alice’s) measurement outcome. Synonymous with “locality loophole”.

2) setting-HV independence: Independence between each measurement setting choice and any hidden variables (HV).

To establish independence of the settings—whether from outcomes (1 above) or from hidden variables (2 above)—the experimentalists employ two strategies:

StrategyA: enforce space-like separation between events that should be independent

StrategyB: appeal to a physical model about the mechanics of the RNG

Of course, there will be LHV models that cannot be excluded in any feasible experimental construction. Among others, superdeterminist models could be responsible for violations of both setting-outcome independence and setting-HV independence, and cannot be excluded through any physical strategies. In lines 636-640, the authors mention specifically that superdeterminist models cannot be excluded, however, there is no mention of other models that can be excluded, such as some where a causal relationship exists between LHVs and setting choices. The goal of a “loophole-free” experiment is to exclude as many LHV models as possible, and represent as thoroughly as possible the categories of LHV models that have been excluded.

To achieve setting-outcome independence: In appendix A of this work, the authors carefully detail how they rely on space-like separation (StrategyA) to convert {locality condition} into {principle of locality, space-like separation} to approach setting-outcome independence using space-like separation.

To achieve setting-HV independence: There is no explicit discussion of how to approach setting-HV

independence. Given the inconsistent definition of “measurement independence” (see above), it’s possible the authors mean lines 671-680 of appendix B as an appeal to StrategyB to achieve setting-HV independence. There is no mention of space-like separation.

These two approaches are inconsistent. If StrategyB is sufficient for claims of independence, then space-like separation would not be necessary to address the locality loophole. Likewise, if space-like separation can help address the locality loophole—which is generally held to be the case, and which the authors argue in Appendix A—then it can also play a role in building setting-HV independence.

This manuscript would benefit significantly from:

- A) A direct discussion about setting-HV independence and the role it plays in this experiment.
- B) An analysis of setting-HV independence from a space-time perspective, indicating which LHV models could be restricted based on the space-time configuration used in this experiment.
- C) An extension of the argument in lines 636-640 to address also the non-superdeterminist models where setting-LHV independence can(/not) be established.

Definition and claims of “loophole-free”:

Consider using “loophole-free” with a little more care:

- Line 66: “loophole-free Bell inequality violations, which do not rely on additional assumptions, were demonstrated in 2015 and the following years...” Ultimately, no test of Bell’s inequality can be truly free of additional assumptions—there’s no such thing as a truly “loophole-free” experiment. We can exclude different and ever larger sets of possible LHV models, but there will always be residual assumptions. Thus the statement that any violations “do not rely on additional assumptions” is too strong.
- This experiment is clearly a tour de force, and that speaks for itself. I would encourage using the word “loophole-free” sparingly—it appears 14 times in the main text and another 11 times in the supplement. When it’s possible, an accurate description of the experiment is more compelling than a shiny buzzword that’s inevitably a stretch. It is in the interest of the authors and the community to communicate about these experiments with meticulousness and care.

Furthermore, with the omission of discussion about setting-HV independence, this experiment glosses over a loophole that has been addressed in other previous experiments. Especially if this manuscript claims to be “loophole-free”, a rigorous discussion of setting-HV independence is in order.

Measurement, readout and reset:

The section on readout is clear and pretty thorough. Still, it would benefit from a statement about the reset fidelity, including the time it takes to achieve. I’d assume the authors wait between trials for long enough for the readout resonator to reset, a statement of the time and performance would be valuable.

The authors point out the presence of a TLS near one of their qubits, which contributes to a lowered readout fidelity. I’m curious if the T1 of this TLS is known. Given the statistical analysis the authors use to close the memory loophole, any possible memory between trials—such as a possible memory mediated by the TLS—could not have an effect on the conclusion, so strictly speaking there’s no need

to include this analysis. I'm just curious. :-)

Line 1785: What is meant by "classical noise"?

Figures:

- Figure1 doesn't do much for me—it's not clear what this figure accomplishes that is not already represented in figure3 or figure4. On a first read, I found the figure distracting because I was looking for more information. Consider sharpening the focus or removing the figure.
- Figure4: consider adding uncertainties for the time values at the far-right of the figure.
- Figure7 is exhaustive. Can it be simplified? For example, it looks like the Alice and Bob setups are almost identical, and there are so many elements that it's difficult to identify similarities and differences. Consider drawing just one analysis setup, and highlight (with description in text and/or figure caption) any element that exists only within Alice or Bob.

Nit-pick questions and suggestions:

- Line 123: "occurring at best at the speed of light" → "occurring at most at the speed of light" or "occurring at the speed of light or slower"
- The text refers a few times to "node C" (e.g. line 903, 908-9, 932, and possibly others), but I don't see "node C" labeled in any diagram.
- The word "qutrit" appears without introduction several times in Appendix H and Appendix I. The manuscript would benefit from a clearer definition of what elements are meant with "qutrit", and how those are distinguished from what is meant with "qubit". For example, line 1415 includes "emitter qubit in the $1/\sqrt{2} (|g\rangle + |f\rangle)$ state", but elsewhere the qubit's states are given as $|g\rangle$ and $|e\rangle$. So should line 1415 reference the qubit, or qutrit? Please review both appendices with this distinction in mind.
- Line 1693-1697: "For experiments using optical photons... the CHSH violation typically is low..." Note that experiments using optical photons closing the fair-sampling loophole typically use the CH-inequality, since the CHSH-inequality assumes fair-sampling while the CH-inequality can be violated only by an experiment showing adequately high collection efficiency.
- Line 1810-1811: "The ultimate length limit of the current setup is given by the length of the laboratory..." This sounds like a current limit, but not an in-principle limit. Since this discussion is focused on how to increase the margins, it would be interesting to hear what in-principle limits exist for this kind of setup.

Author Rebuttals to Initial Comments:

Reply to the referees

Note from the authors:

In order to adapt our publication better to the format expected by Nature, we have split the revised version of the paper into two separate documents, one containing the main text and the other the supplementary information.

In the following document we show the referee's comments in black and display our responses in blue.

Referee #1 (Remarks to the Author):

Summary. The paper performs a loophole-free Bell experiment with entangled qubits realized by superconducting circuits separated by 30 meters.

Originality and significance. Loophole-free Bell experiments are exceptionally challenging technically, requiring long-distance high-quality entanglement of qubits with ultra-fast measuring schemes that can toggle between different measurement angles in timeframes measured in nanoseconds. Accordingly, the first such experiments were not performed until 2015 and no more than a handful of experimental groups have achieved the loophole-free benchmark to date. Loophole-free Bell experiments are important because only they can definitively lay claim to exhibit nonlocality as a physical phenomenon independent of any assumptions about the correctness of quantum mechanics and/or the quantum modeling of the experiment, and they are required for the execution of the most secure device-independent quantum communication protocols. The 2022 Nobel Prize was awarded largely for, in essence, early experimental work towards the goal of loophole-free Bell tests.

The current experiment is notable in being the first loophole free Bell experiment employing superconducting circuits. (Previous experiments had used photons, NV centers, and neutral atoms). While demonstrating a loophole-free Bell experiment in any new platform is a major scientific achievement, this experiment can lay special claim to additional significance of having done this with the same platform as used for quantum supremacy experiments reported in Ref. [10] and [Phys. Rev. Lett. 127:180501 (2021)]. Thus the current experiment shows that 1) a quantum advantage outperforming classical computers (per the above references, modulo any potential critiques/re-assessments for which I'm not up-to-date) can be complemented with 2) a quantum advantage in exhibiting nonlocal correlations -- arguably THE two main superclassical quantum science capabilities of the second quantum revolution -- all in one system of superconducting qubits. I think this is

fascinating, not just because of the potential for applications towards implementing integrated quantum information protocols, but on a more pure science level of stimulating thought on the nature of quantum advantages and the not-completely-understood link between the two types (computation and nonlocality). Indeed I believe the manuscript could have done a bit of a better job alluding to this deeper exciting significance along the lines of what I am loosely trying to get at here.

In light of this, I recommend this manuscript for publication in Nature, provided the authors can satisfactorily address some mostly minor clarifications that I ask about below.

Data & methodology. The overall scientific quality is high. The authors rightly recognize the importance of precise timing. I do have a few requests for clarification regarding this.

First, the start event: The start event is characterized as the time the "laser goes above threshold" (line 808); to confirm in layman's terms, is this as the time when (or at least no later than when) the laser starts emitting the photons that will then be measured to obtain the randomness for settings?

We thank the referee for this question. The referee's interpretation is correct: when the laser "goes above threshold" it begins to emit photons that will be used to obtain the random output, and this is the relevant time for determining the "creation time" of the generated randomness.

To give a fuller picture, we now describe this aspect in more detail. What we write here is a summary of descriptions published in [Abellan et al. PRL 2015] (already cited in the original manuscript as reference 48) and [Abellan et al. Optics Express 2014], which we now cite in the revised manuscript. As already described in the original manuscript, the laser is periodically taken below and above the lasing threshold by a square-wave modulation of the laser's injection current. Optically, the threshold condition is this: the net round-trip gain (determined by the density of free charge carriers in the gain region) equals the round-trip cavity losses (determined by the fixed cavity geometry, especially the transmission of the output coupler). Taking the injection current above a certain level thus takes the laser above threshold. We use the rising edge of the input current square wave, an electronic signal that can be precisely measured, as an indicator of when the laser crosses threshold (more on this below). This process is described in greater detail in [Abellan et al. Optics Express 2014].

We note that this method of estimating the creation time is conservative, in that it does not include two small (~ 0.1 ns) delays between the electronic rising edge and the end of the strong phase diffusion. These are the ~ 100 ps lifetime of the injected carriers and the ~ 100 ps time after crossing threshold for the photon number to rise to its equilibrium value. If these were included, the duration of the randomness generation process would be slightly less than the 17.1 ns we report, because the randomness creation time would be shortly after the electronic rising edge that we are using as a marker. This would make space-like separation slightly easier to achieve, as one can appreciate by looking at Fig. 4, and imagining that the start time (the stars) are moved later (upward in the space-time diagram), while leaving all the other space-time events unchanged.

And since this time can't be measured directly during runtime, how exactly do you obtain the 17.10+- .14 ns figure (807) for the delay until the observable reference event t_0 ? The discussion

(812-815) suggests a characterization of the internals of the device, which may be OK, but a little more information/clarity would be helpful here

As described above, the threshold condition is an electronic condition as much as it is an optical condition, and its timing can be characterized by electronic measurements. We measure, using a fast oscilloscope, the time delays between the laser current rising edge and other events within the signal chain. We note that the number reported, 17.10 ns, represents a single time delay measurement, the delay from the rising edge of the laser current to the rising edge of the RNG output voltage. This net delay is the only RNG delay that is relevant to the time budget of the experiment. We report its division into propagation delay, conversion delay, and other delays as additional information that may be of interest to the expert reader, but these do not enter into the timing budget as they are all subsumed in the directly measured 17.10 ns delay.

To clarify we have expanded the discussion in the relevant paragraph (lines 352-376 in the revised manuscript [Supplementary Information]).

And can the authors address how this choice of start event compares to the choice of start event for the other LHF tests, just as they did a comparison for the choice of end event - was the start event the same as for these other experiments (which used similar/same RNGs)?

Indeed, the start event was the same in other experiments using phase-diffusion RNGs, as was the methodology for measuring time delays.

To clarify, we have added a sentence to the relevant paragraph (lines 369-371 in the revised manuscript [Supplementary Information]).

Second, concerning the establishment of a common laboratory reference frame (898-899). This is of key importance; when determining the Bell time budget at Alice, you need to measure the difference of time coordinates between 2 events, the latter of which is defined as the time a specific spatial location at Alice enters the forward light cone of an event that occurs 25 meters away (at Bob). Thus, to measure the duration of this window, you need a very precise synchronization of clocks at Alice and Bob to sub-nanosecond precision. The authors say that this is done by "sending a square pulse from a central trigger device" (900) to the other devices, then enter into a long terminology-filled discussion that I don't have the technical expertise to follow. I suspect this took more effort than (just) making sure "output signals arrive simultaneously through equally long [12.5 m?] cables" (right?). Overall I am willing to defer to the experimental expertise of the team, if they can

re-affirm in the reply that they are capable of synchronizing clocks separated by 25m to within fractions of a nanosecond with super-high confidence, and that when measuring durations of various steps (fig 9) that they are carefully accounting for how long various signals take to travel (1 m \geq 3 ns!) from the thing whose duration they are measuring to whatever nearest clock that is keeping time (I think the three oscilloscopes in 1142?). I will state that the descriptions of measurement procedures that I can better understand and follow (for instance Appendix E.2

measuring distance from Alice to Bob) meticulously took into account everything appropriately and made some sound conservative choices.

The accuracy of the timing of the overall experiment is assured in two essential steps. First, we assert that the local clocks at every instrument at any given physical location of the experiment (e.g. at A and B) tick at the same rate. This is enabled by sharing a phase stable reference clock signal between all instruments, which is used at every instrument to run a phase-locked loop, thereby establishing a common clock. (Since some of the instruments have different requirements for their internal clocks, this procedure requires some care, as discussed in detail in Appendix V of the revised manuscript). In a second step, the absolute timing between different instruments is measured by receiving the output signals which the instruments create (already having a synchronized clock ticking at the same rate) at a common set of oscilloscopes, where given the known physical cable lengths, the runtime of signals can be accurately determined. In these two steps, it is assured that the clocks all tick at the same rate and that the relative time at which signals are available at the relevant locations in the experimental setup are known with the required accuracy.

If the referee is interested in more details, in the following we provide a more detailed explanation of such a timing measurement.

In particular, we demonstrate how to experimentally determine the synchronization offset between the two setups ($t_0' - t_0 = 0.3 \pm 0.07$ ns; see line 1485 and Figure S12 in the revised manuscript [Supplementary Information]). For this measurement we use two approximately 20 m long cables which connect the main AWGs (Tektronix 5014) at the two outer nodes to a central timing oscilloscope in the middle of the link (Chronos), see Figure S5 in the revised manuscript.

These two cables do not have exactly the same length. Therefore, in a first step, we determine the difference in propagation delay of a signal going through the two cables. As shown in the attached Figure below, for this measurement an AWG5014 generates a square pulse, which we split and send through both cables to detect their relative arrival times using a single oscilloscope. We measure the difference in the arrival time of the rising edge of the two signals on the oscilloscope to be $t_{\text{delay,cables}} = 27.327 \pm 0.051$ ns. For this measurement we use the built-in measurement tools of the oscilloscope which automatically determine the time delay of two rising edges at the 50% amplitude threshold level. The measurement is based on approximately 5000 shots, which determines its standard deviation (see lower part of the screenshot below). We verify that the result remains the same when swapping the oscilloscope input channels of the two cables.

In the actual Bell test, we then connect these two cables between the AWG5014 at the two nodes and the central timing scope, see Figure S5 in the manuscript. During the Bell test we constantly monitor the arrival time of square pulses generated by the remote AWG5014s at the central timing oscilloscope. We detect the difference in arrival time of these two signals in the same manner as before to be $t_{\text{delay, experiment}} = 27.609 \pm 0.033$ ns. We therefore get $t_{\text{sync offset}} = t_{\text{delay, experiment}} - t_{\text{delay, cables}} = 0.282 \pm 0.061$ ns using Gaussian error propagation on the sigma. We conservatively round up this result to $t_{\text{sync offset}} = 0.3 \pm 0.07$ ns, which is the value we refer to in the publication as $t_0' - t_0$.

In a similar manner, we measure the other parts of the protocol duration described in Appendix XI.

Third, there are aspects of photon generation/detection that surprised me and I want to ask about. This request is just a sanity check on my own part and does not require changes in the paper if the authors feel these aspects would be unsurprising to a specialist, which I would suspect is the case. So: I understood from the manuscript that prior to every trial, to create entanglement, a single microwave photon is emitted from one party to the other along the 25m long waveguide (490, 1387). This is a little surprising to me because I thought it was hard to create single photon on demand sources: I'm familiar with low-power (optical) lasers in the LHF bell tests Ref [6] and [7] where if the low-power laser was emitting on average one photon during a time unit you would have a Poisson distribution leading to a large probability of zero-, two-, three- photon events with the higher-number events fairly frequent (maybe 30% probability) and especially detrimental to the Bell test. If I

understand (1461-1463) correctly, higher-number events are not unknown to your setup but these must be happening at much lower probability. So is your setup just fundamentally different where on-demand single-photon generation is not difficult? On a related note, it is my understanding that single photon detection is hard, but it seems like your apparatus for creating remote entanglement effectively detects the incoming photon at B with very high efficiency (1398), or more precisely could be converted into a very efficient photon detector with perhaps different choice of measurement if that was the goal instead of creating entanglement. Can you comment on this as well? It seems to me like possible answers are a) microwave is different than optical frequency, 2) waveguides (or ultra-cold waveguides) are different than optical fiber and/or free space 3) lasers are different than

your photon source 4) actually, it's not just one photon traveling from Alice to Bob prior to each trial to create the entanglement, it's a whole bunch.

Indeed, in our experiment, the emitter qubit (at site A) is first entangled with a single propagating photon in an emission process induced by a coherent drive. At the receiver qubit (at site B) this photon is absorbed with close to unit efficiency, to transfer the qubit-photon entanglement to qubit-qubit entanglement (between site A and B). This was first demonstrated by our lab in [Kurpiers et al., Nature 558, 264-267 (2018), Ref. 12 in the revised manuscript]. The process is based on a proposal by Cirac et al. [new Ref. 57 in the revised manuscript] and is discussed in detail in our Nature paper (Ref. 12). So, the referee is correct in pointing out that the entanglement is realized by transferring a single photon (instead of a weak coherent tone) and that, indeed, at the absorber this process could be used for single photon detection. Efficient single photon detection has been realized for microwave photons with superconducting circuits, see e.g. [Besse et al., PRX 8, 021003 (2018)] and references therein. For the benefit of the reader, we have improved the Methods section which discusses the generation of entanglement.

Appropriate use of statistics and treatment of uncertainties. The statistical method detailed in Appendix L is appropriate and the computed p-value 10^{-108} looks consistent with the appropriate figure (L1), which is the (extremely small) probability of a Binomial random variable with $p(\text{success})=0.75$ exceeding 796228 successes in 2^{20} trials (a 0.75934 success rate). Putting error bars on the CHSH violation is fine when considering this as a measurement of a physical parameter but statements using this uncertainty bar to quantify violating Bell's inequality ("by 22 standard deviations" in abstract, see also line 339) suggest this as a stand-in for a p-value for violating locality, which is incorrect. While mostly harmless, I recommend removing these statements.

We agree that this statement may be misleading in the context Referee 1 mentions. We have therefore removed the statement "*violating Bell's inequality by more than 22 standard deviations,*" in the abstract and directly refer to the p-value. At the same time, we think that for an experimental result it is still important and useful to quote the S-value together with its standard deviation. We therefore prefer to keep the numeric value of the standard deviation ($+0.0033$) on the S-value in the abstract.

On a related note, the use of standard deviations in lines 1715-1723 as a sort of metric to measure the absolute size of Bell violations doesn't make sense - what matters in this context is comparing the absolute sizes (eg, 2.6 is better than 2.4) while acknowledging the precision of each figure; but 2.4 with better precision (possibly just meaning more statistics were taken) isn't suddenly better than 2.6 because it is more standard deviations above 2.

While we think the precision is an important measure, we overall agree that this type of phrasing may surprise some readers. Since it was not the intention in this paragraph to focus on the precisions, but rather on comparing the full experimental results consisting of the S-value and its precision, we have here removed the explicit statements on the statistical significance of the listed experiments. It now reads: "*(...) Early Bell tests using the CHSH [CHSH1969] inequality, such as [Aspect1982], invoked the fair sampling assumption and found an S-value of $S=2.697\pm 0.015$. The first experiment to close the locality loophole also relied on the fair sampling assumption and*

measured an S-value of $S=2.73\pm 0.02$ [Weihs1998]. The first experiment to avoid this assumption by closing the detection loophole while not addressing other loopholes reached $S=2.25\pm 0.03$ [Rowe2001]."

Note that we have added a new Methods section to the revised manuscript (lines 642-664 [Main text]) in which we discuss the difference between the standard deviation and the p-value as measures for the statistical significance of the result, in order to address a related comment made by Referee 2.

Conclusions. Overall, exceptional. I do have some concerns about how the "principle of locality" is presented, but these can be easily fixed with a better discussion. Specifically: the statement that the principle of locality "derives from the observation that no causal influence can propagate faster than the speed of light" (38-39, see related at 55-56) may be misinterpreted to suggest that Bell violations imply causal influences faster than light. Indeed the "principle of locality" is not term with a standard well-accepted definition in the literature, and the proffered definition in Appendix A of "allowing events to be influenced by an arbitrary action in their past light-cone" (624) is faulty (surely QM "allows" this). A better definition, perhaps rooted in local hidden variables terminology, should be formulated. Note, I agree with the authors that "realism" is a term that is perfectly fine to avoid.

We thank the Referee for their careful reading of our work and for pointing out potential misinterpretations. To avoid any misunderstanding, we changed the terminology of 'principle of locality' to adopt the standard terminology of 'local causality' introduced by J. S. Bell himself, see e.g. [J. S. Bell, in *Speakable and Unsayable in Quantum Mechanics*, second edition, pp 232-248 (2004)]. Accordingly, the definition given in the main text now states that the "principle of local causality"

"derives from the expectation that the causes of an event are to be found in its neighborhood.",

which involves no notion of 'influence' anymore. The definition in the appendix has also been modified to describing models

"only allowing an event to depend on the content of its past light cone."

Additionally, we expanded the discussion in the corresponding Appendix to emphasize the role of local hidden variable models (LHVMS) in the testing of theories satisfying the principle of local causality.

Other things: The locality loophole is opened when relevant events aren't outside light cones, not the statement (604-605) which would incorrectly suggest that QM exploits the locality loophole to violate Bell inequalities (note A2 is violated by QM while A3 is obeyed by QM).

We thank the Referee for this remark. We agree that the informal description of the locality loophole in Appendix A was problematic. We have modified the sentence as follows:

“In a space-time configuration allowing for communication from A to B, A's setting choice a may become available at B's location, in which case the statistics produced by a hidden variable model may not satisfy Eq. (A2), thus opening the locality loophole.”

Similarly, lines 609-615 is wrong; space like separation of the events doesn't "guarantee the validity of A2"; see again QM.

The whole paragraph containing this sentence has been reworded to better describe how space-like separation allows to test the principle of local causality, as summarized in Eq. (A12).

Finally, I believe that one can claim that conclusions not resting on discussions of realism are clearer, more precise, or better posed (etc.) but I would hesitate to say "stronger" (665) because the fuzziness of the realism concept is the problem, not that there is some clearly defined mathematical definition of realism (something like A1 and A2) that is in fact superfluous.

We thank the Referee for sharing his impression on this question. Following their recommendation, we have modified the sentence to (lines 206-208 [Supplementary Information])

“However, by not leaving non-realism as an alternative to non-locality, our conclusion is clearer.”

Suggested improvements. In addition to what I have discussed above, I have the following small edits:

(91-92) "making bit strings more random, a task impossible by classical means" doesn't sound right, maybe reword or just stop sentence at "randomness amplification" if a better description this concept about the "bit strings" (S-V sources) gets too digressive;

We agree that the terminology using bit strings may not be optimal. We have changed the phrasing as follows: *“Further applications of Bell tests include (...) and randomness amplification, improving the quality of a source of randomness in a certified manner [Colbeck2012,Kessler2017], a task impossible to achieve by purely classical means.”*

suggest writing either $t_{\text{star}}=0$ at (116) or $t - t_{\text{star}} < td=d/c$ in (124);

We have adapted this in the main text. The corresponding line 119 [Main text] now reads *“(...) an individual trial of a Bell test begins at time $t_{\text{star}}=0$ (...)”*

I suggest depicting and labeling "d" (from line 122) in Figure 1;

We have added a label of the distance d in Figure 1.

I see $|g\rangle$ and $|e\rangle$ defined as ground and excited in (139) but is $|f\rangle$ defined anywhere;

$|f\rangle$ denotes the second excited level of the quantum system. We see that an explicit definition was missing in the original manuscript, and we have therefore added the definition of $|f\rangle$ in the Methods Section “Superconducting Circuits” where $|f\rangle$ is mentioned for the first time (line 515 in the revised version [Main text]).

strictly speaking this does not require addressing if obvious to a specialist but I am just curious why it is required for the entire 25m waveguide to be so cold (199-201) when optical fibers can transmit photons without too much loss at room temperature;

The quantum circuits and the connections among them need to be cooled to temperatures where the typical energy of thermal fluctuations $k_B T$ is much smaller than the energy of the transition between the lowest two energy states of the qubit, $\hbar \omega_{ge}$. For superconducting circuits operating at frequencies of a few GHz, i.e. in the microwave domain, the operating temperature must be on the order of 20 mK correspondingly. This not only holds for the static quantum circuits, but also for the connections between them, such as a rectangular waveguide. It ensures that these channels add minimal thermal noise to the microwave fields it carries. Furthermore, the loss of these waveguides is minimized when they become superconducting, i.e. at $T < 1.2\text{K}$ for our aluminum waveguide.

We have added a Methods section on superconducting circuits (lines 499-525 in the revised manuscript [Main text]) which briefly addresses this aspect, as motivated by the referee's suggestion found below.

around (343) I would re-mention that the statistical analysis leading to the p-value of 10^{-108} is robust to memory effects and cite appendix L and the references therein for the methods used to perform said statistical analysis and obtain the figure;

At the end of this paragraph, we have added the statement: *"The statistical method used here is robust to memory effects, see Supplementary Section X."* to the revised manuscript (lines 354-355 in the revised manuscript [Main text]).

it is not quite correct to state that high Bell violations are necessary for device-independent protocols (406-409): this is true for DI-quantum key distribution for which there is a threshold strictly larger than 2, but not true for DI-random number generation where any violation above 2 can be used;

We agree with that statement and that while very useful, it is not strictly necessary for some of the device-independent protocols to have a high S-value. In the lines mentioned by Referee 1 we have replaced the phrasing *"it is essential"* with *"it is desirable"* to avoid that the reader could interpret our statement in a too strict sense. Note that we have implemented similar adjustment in Supplementary Section VIII of the revised manuscript (line 937).

(A5) remove last "x" from conditioner; define angled brackets notation in (A7);

We have addressed both points in the revised manuscript.

I appreciate that a sanity check was performed for the equiprobability of settings frequencies (774) and a further sanity check to ensure no signaling (independence of Alice's outcome probabilities from Bob's setting choice and vice versa) would be nice to report as was done with at least some of the other LHF experiments;

We agree that an analysis of whether the no-signaling conditions are fulfilled is helpful. We have therefore added a subsection to Appendix X of the revised manuscript. In a first, traditional approach, we perform a pooled two-proportion z-test on the data, and we find p-values which suggest that the null hypothesis of no-signaling does not have to be rejected.

We also analyzed the no-signaling character of the experimental data with the help of the prediction-based ratio (PBR) method as described in [Y-C. Liang, Y. Zhang, Entropy 21, 185 (2019)]. This analysis didn't reveal any significant trace of signaling either.

Note that this is expected to also address a corresponding comment of Referee 2.

"copper" is capitalized multiple times in the SM;

We have corrected these mistakes in the revised manuscript.

I can guess at the reason but state why signals are attenuated as they travel into the cold regions (865-866);

We have added a sentence at the end of the corresponding paragraph to clarify this aspect (lines 584-588 in the revised manuscript [Supplementary Information]): *"The signal generated by the room temperature electronics co-propagates with excess noise and thermal radiation along the signal lines and is attenuated and re-thermalized in the attenuators installed in the experimental wiring, see Ref. [Krinner2019] for details."*

align labeling of "Node C" in appendix D and "Chronos" in Fig 7;

In the first paragraph of Appendix V.B we have clarified in the revised manuscript what the labels of the nodes A, B and C stand for.

in Appendix E.2 fix the many terminology problems - be clear that Bell distance is in fact a time interval in the laboratory reference frame, the norm notation in (990) is mathematically inappropriate, be clear whether the p-vectors are locations or events because you are measuring the physical distance between them in (1003), and a small thing but is there in fact just one additional neglected spatial direction not two (1023) as you are projecting onto the the 2-D floor not a 1-D line;

We thank the Referee for carefully reading this part. In the revised manuscript, we now use standard notation, and we have made clear that the vectors represent spatial positions. To reflect this better, we have also changed the name of the vector from p to r .

We note that it is in fact two spatial directions that we neglect: we do not consider the z-axis, and we also neglect the exact position of r_{start} and r_{end} in the y-direction. This is a conservative

choice, as it effectively shortens the Bell distance with respect to the actual physical distance between the two points. To clarify that the y-component is not of relevance in the laser measurement we have modified the following sentences in the revised manuscript (lines 1397-1405 [Supplementary Information]): *“For this measurement, using a cross line laser (Leica Lino L2P5), we first orthogonally project r_{start} and r_{end} onto a the edge between the floor and the wall of the laboratory (see “marker” label in Fig. S11b) as the direct line of sight between the start and stop points is blocked by the cryogenic instrumentation. We ensured that the projection is orthogonal by verifying that $|d-d'| \leq 5\text{mm}$ over the distance of the whole 30-meter-long system (Fig.S11b)”*.

citation of [50] in (1703) seems to be an error.

We thank the Referee for noticing this mistake. We have replaced the citation with the intended one throughout the whole paper: *J. Barrett et al., PRL 95, 010503 (2005)*

References. The reference list is comprehensive and the location of citations is overall good with only minor recommended changes mentioned above.

Clarity and context. Generally very clear, and good context is provided. I have one recommendation, that the authors state what IS a superconducting qubit - this was strangely absent from Ref. [10] as well. Perhaps this is technical and hard to describe, but at a minimum, what is the degree of freedom that can exist at two levels to implement a qubit? Also, there is talk of a qutrit that is somehow coupled to the qubit playing some role in sending and receiving photons to create entanglement and/or read out measurement outcomes, I'm curious to hear a little more about this mechanism as well without having to consult references. A full description of these things is obviously beyond the scope of the present manuscript, hopefully there is room for perhaps a phrase or two in the main manuscript stating what the mechanisms are, and maybe a little more in the SM with indications where the mechanisms are located in Fig. 12.

To address the comment, we have added a Methods section to the revised manuscript which provides a short basic discussion of superconducting qubits (lines 499-525 [Main text]).

Referee #2 (Remarks to the Author):

This work reports a loophole-free Bell test with quantum measurements on superconducting qubits. To achieve this goal, the authors entangle a pair of superconducting qubits separated by a physical distance of 30 meters. This physical distance is much larger than those between two superconducting qubits achieved in previous works. At the same time, the two remote qubits reported here are evaluated to be strongly entangled such that a significant violation of the CHSH Bell inequality can be observed accordingly. Moreover, the authors achieve rapid, high-fidelity readout of superconducting qubit states using the state-of-the-art readout scheme optimized in their previous work [Ref. 12]. These achievements enable the authors to close the locality and the detection loopholes for the first time in a Bell test using superconducting qubits. In my personal opinion, the loophole-free Bell test reported in this work is a landmark experiment.

The manuscript provides a good description of most aspects of this work. Nevertheless, there are a few statements which I found confusing or unclear:

1) In the main text and Appendix A, the authors specify the two underlying assumptions in the derivation of Bell inequalities --- locality and measurement independence. I cannot fully agree with this statement. The use of local hidden variables to describe a quantum experiment implies that the measurement outcomes under different settings can coexist or be predetermined before the measurements. To my understanding, the coexistence or predetermination is not captured by the notion of locality but by the notion of realism in the literature. In fact, as shown by Fine [Phys. Rev. Lett. 48, 291 (1982)], probabilistic, local hidden variable models and deterministic hidden variable models are equally powerful. Considering the above, I prefer to say that the two assumptions underlying the derivation of Bell inequalities are local realism and measurement independence.

We thank the Referee for carefully reading our publication and for sharing their personal view on the interpretation of Bell inequality violations. This, together with other related comments, have

triggered us to revisit some of the sections in Appendix A to describe our approach in a clearer manner.

We acknowledge that discussing Bell tests in the context of local realism, which is the way the Referee prefers, is a commonly used approach, and it is a natural one. It is not the only possible approach, however, as we describe in the main text and in Appendix A. As shown there, it is possible to define local hidden variable models (LHVMs) and to derive Bell inequalities without relying on a notion of realism.

To be more precise, we do consider the outcomes observed during the experiment as corresponding to something 'real', but we see no reason to formulate statements on the hypothetical existence of outcomes before the measurements are performed. In turn, no discussion on the pre-existence of measurement results is required in our derivation of Bell inequalities, which we think is valuable.

Other works have made alternative choices and derived Bell inequalities under different assumptions, notably under the combined assumption of locality and 'realism' (completed by measurement independence), usually referred to as 'local realism' as mentioned by the Referee. It is well-known that identical results can sometimes be obtained starting from different sets of assumptions. This also applies to Bell inequalities, and presents no contradiction, see e.g. H. M. Wiseman, *J. of Phys. A: Math. and Theor.* 47, 424001 (2014). All of these facts are recalled in the end of Appendix A. Given the rather philosophical character of the realism assumption, as opposed to being a physical/testable assumption (by definition, a result cannot be observed before it is measured) we see it as a benefit to deliver a conclusion free of this notion, but other people may have different personal preferences.

We would like to point out that our approach is far from isolated. Numerous researchers in the field have questioned the realism assumption in the context of Bell's theorem and derived versions of the theorem exempt of it. For instance, in *Foundations of Physics* 44, 736 (2014), P. Bierhorst describes the main problem with the term realism as follows: "Sometimes it is claimed that it is not locality, but realism that must be abandoned. However, there is some debate about whether realism is a well-defined, required concept in the context of Bell experiments, and there is no clear invocation of realism at any point in this paper. (...) To claim that the CHSH inequality rests on an assumption of realism requires being able to identify which of the assumptions (...) should be identified with realism."

A particularly strong advocate against the terminology of realism is also T. Norsen, as argued in *Foundations of Physics* 37, 311–340 (2007): "I will conclude by pleading with the physics community to revisit these crucial foundational issues. We must reject the thoughtless and confused use of terminology such as 'local realism' — and all of the misunderstandings on which this terminology rests, and which the terminology, in turn, helps perpetuate."

Third, F. Laudisa concludes in his historical analysis in <https://arxiv.org/abs/2205.05452>: "Seen in retrospect, the history of the debates on the foundational implications of the [Bell's] theorem displayed very soon an attempt to put the theorem in a perspective that was not entirely motivated by its very assumptions: the history itself of quantum theory appeared to be affected heavily by the issue of what the fate of a not-well-specified 'realism' would have been, so heavily that, already in the temporal surroundings of the Bell theorem publication, the interpretation of its general meaning could not avoid to fall into the intense gravitational field of the 'local-realistic' narrative."

Most notably, our approach follows John S. Bell's very own efforts to purify the assumptions of his celebrated theorem. A good introduction on the question can be found in his article "La nouvelle cuisine", our Ref. [3], where Bell also uses the terminology of local causality to conclude "The obvious definition of 'local causality' does not work in quantum mechanics, and this cannot be attributed to the incompleteness of that theory."

Finally, we note that Referee #1 agrees with us that no notion of realism is needed to define a Bell inequality.

We again thank the referee for sharing their thoughts on this aspect and we believe the corresponding changes in Appendix A have helped to improve the publication. Their comments were useful in the process of rewording the description of our approach in the paper in a clearer way.

In addition, I am not sure how to interpret the implication relation stated in Eq. (A9). To me, the meanings of locality and space-like separation overlaps.

Locality is the principle that events can only depend on the content of their past light-cone. A theory or model may follow this principle or not. Space-like separation, on the other hand, is not a principle satisfied by a model, but a condition that applies to events in space-time: some events in space-time are space-like separated, while others are not. These two conditions apply to different objects and are thus of very different nature. In fact, they can be combined in all possible manners: it is possible to consider events that are space-like separated in a theory that does not satisfy the locality principle for instance, or time-like separated events in a theory that does satisfy the locality principle, etc.

Eq. (A9) (Eq. 12 in the revised manuscript) is an implication in the case where the conjunction of the three conditions listed apply. Hence, it is concerned with the prediction of a theory satisfying the locality principle in a situation in which events are space-like separated. The claim is that the predictions of such a theory in this situation must satisfy Bell's inequalities when the measurement independence condition is fulfilled.

Note that our definition of space-like separation matches text-book special relativity. Similarly, our definition of locality matches the one used in the literature, as can be found in John Bell's paper mentioned above under the name of local causality that we use now, to cite just one reference.

2) In the Methods section, it is mentioned that "the duration of the pulse sequence (for entanglement creation) is irrelevant for closing the locality loophole as it occurs before the basis choice is initiated ...". However, I didn't catch a statement in the current manuscript on at which time point the entanglement is nominally created.

We thank the Referee for this comment. In the Methods section of the revised manuscript we have added a statement clarifying this aspect (lines 538-541 [Main text]).

A related question is whether the creation of entanglement and the determination of basis choices can be space-like separated in the experiment reported here. I understand that it may be difficult to

achieve such space-like separation in the current experiment, as neither a heralded method nor a middle station between Alice and Bob is used for entanglement creation/distribution. However, the readers may still be interested in seeing the comparison between the loophole-free Bell test reported here and those reported in previous works.

We thank the Referee for this comment. We agree that readers may be interested in the suggested comparison, and we have added such a comparison to Methods: Random Basis Selection and Measurement Independence (lines 593-641 in the revised manuscript [Main text])

3) The description of the entanglement-creation scheme in the Methods section is brief. The working of the scheme would be clear if the joint quantum state of Alice's qubit, Bob's qubit and the intermediate photon could be tracked after each step.

We hope that this aspect becomes clearer with the new wording in the revised methods section. For the benefit of the referee, we provide here a more detailed explanation:

We generate the Bell state $|\psi^+\rangle = (|ge\rangle + |eg\rangle)/\sqrt{2}$ between the two remote qubits using the scheme described in [Cirac1997] and [Kurpiers2018]. First, we prepare qubit A in state $(|e\rangle + |f\rangle)/\sqrt{2}$. In combination with B, the joint quantum state reads $(|e,0,g,0\rangle + |f,0,g,0\rangle)/\sqrt{2}$ where the first two indices of a ket vector denote the state of qutrit A ($|g\rangle$ is the ground state, and $|e\rangle$ and $|f\rangle$ the first and second excited levels) and the Fock state of the coupled transfer resonator. The last two elements denote the state of the qutrit and transfer resonator at B, respectively.

Next, we drive the $|f_0\rangle \leftrightarrow |g_1\rangle$ transition of the coupled qubit-transfer resonator system at A to transfer the excitation in the f-level into a photon in the coupled resonator, which yields the total state $(|e,0,g,0\rangle + |g,1,g,0\rangle)/\sqrt{2}$. The photon in transfer resonator A decays into the waveguide at rate κ_A (see Appendices IV and VI), travels to the other node of the quantum link, and enters the transfer resonator there with rate κ_B , yielding state $(|e,0,g,0\rangle + |g,0,g,1\rangle)/\sqrt{2}$. At node B, we reabsorb the photon again using the local $|f_0\rangle \leftrightarrow |g_1\rangle$ transition, which transforms the joint state into $(|e,0,g,0\rangle + |g,0,f,0\rangle)/\sqrt{2}$. Finally, we map the f-level at B to the g-e manifold using a π rotation, yielding the final two-qubit Bell state $|\psi^+\rangle = (|e,0,g,0\rangle + |g,0,e,0\rangle)/\sqrt{2}$.

4) In Appendix I, terms like transmission fidelity, transfer efficiency, and transfer fidelity are used. Are they referring to the same thing? If yes, it would be better to keep using one of these terms.

In the revised manuscript we only use the term "transfer efficiency".

5) In Fig. 14, it is mentioned that "the optimal integration weights are shown in dark blue". However, it is not clear to me in what sense the integration weights are optimal. In addition, I wonder whether the integration weights and the state assignment thresholds are fixed before analyzing the Bell-test data.

We have added a clarifying statement in the revised Appendix (lines 849-854 of the revised manuscript [Supplementary Information]): *The signal-to-noise ratio scales with the separation of the*

two readout traces under the assumption that all points are exposed to Gaussian white noise [Gambetta et al., PRA 76, 012325 (2007)]. The presented readout scheme therefore optimizes the integration in the sense that the weighting favors data points with higher signal-to-noise ratio."

In fact, the integration weights and the state assignment thresholds are fixed before the Bell test experiments are performed. We have also added a statement to clarify this aspect in the revised Appendix (lines 856-858).

6) The definition of a p-value needs some work. A p-value is the probability, if the null hypothesis (for example, local realism) holds, of yielding a statistic as extreme as the observed one. Note that it is not necessarily equal to the probability of yielding the observed statistics according to the null. Given a Bell-test configuration, there are many different local hidden variable models. In this case, the p-value is defined as the worst-case tail probability. In view of the above, Eq. (L1) and the sentences around it need to be clarified. Several variables in Eq. (L1), for example c and p_{win} , also need to be explained. Side comment: the upper bound on the p-value obtained in Eq. (L6) is comparable to that according to Theorem 2 (and Corollary 8) in arXiv:1709.04078 by Wills et al.

In the revised manuscript, we have changed the sentence introducing Eq. (19) as follows (line numbers 1239-1241 [Supplementary Information]): *"The probability, of the p-value of a local hidden variable model yielding statistics at least as extreme as the observed ones is given by (...)"*. Note that we have also modified the first sentence of the same section (line numbers 1218-1220 in the revised manuscript) in the same manner: *"The p-value for the Bell test is given by the probability of a local hidden variable model reproducing statistics at least as extreme as the results of our experiment."* The variables in Eq. (19) are already sufficiently well defined in our opinion. We define them in the sentence which follows in the original manuscript, just below (19), where we state that C^i_n are binomial coefficients and p_{win} is the success probability of a single trial. p_{win} was explained in the first paragraph of the section.

7) In the end of Appendix K, it is mentioned that "from this data (in Table IV) we can calculate the CHSH S-value and the corresponding statistics". However, there seems to be no details behind in the current manuscript.

In the revised manuscript, we have merged appendices K and L into a single appendix (new labelling: Additional Information Section X). We believe that this change improves the logic here, and as a consequence we have removed the sentence stated above.

A related comment lies in the evaluation of experimental evidence against local realism. From the discussion in Appendix M.1, one can see that the Bell tests performed at the early stage use the number of standard deviations, while the recent loophole-free Bell tests summarize their evidence against local realism into p-values. It may be helpful to point out this and briefly explain the underlying reason, as general readers may wonder why different metrics are used. Note that the potential problems with the number of standard deviations have been discussed in previous works, see Ref. [4] and Phys. Rev. A 84, 062118 (2011) by Zhang, Glancy, and Knill.

We agree that having a short discussion on that aspect is useful. We have therefore, in the revised manuscript, added a Methods section (lines 642-664 in the revised manuscript [Main text]) which discussed the use of the standard deviation and the p-value as a metric for the statistical significance of the result. We also have added a reference to Ref. [5] (of the revised manuscript) and to Zhang et al., Phys. Rev. A 84, 062118 (2011).

I also have a couple of additional suggestions and observe several minor problems as follows:

1) It could be helpful to testify whether the experimental data presented in Table IV satisfies no-signaling conditions.

We agree that an analysis of whether the no-signaling conditions are fulfilled is helpful. We have therefore added a corresponding subsection to Appendix X of the revised manuscript. In a first, traditional approach, we perform a pooled two-proportion z-test on the data, and we find p-values which suggest that the null hypothesis of no-signaling does not have to be rejected.

We also analyzed the no-signaling character of the experimental data with the help of the prediction-based ratio (PBR) method as described in the recent publication [Y-C. Liang, Y. Zhang, Entropy 21, 185 (2019)]. This analysis didn't reveal any significant trace of signaling either.

2) In Fig. 2(a): It might be better to label the readout fidelity and concurrence without correction as these quantities directly determine the Bell violation observed in the experiment. Is my understanding correct?

The observed Bell violation is determined by the readout fidelities and by the concurrence of the Bell state, see Eq.1 in the main text. This approach is useful because it allows to disentangle the two main sources of infidelity. It means that first of all the S-value depends on the concurrence of the Bell state (which we consider to be able to measure with perfect fidelity, using readout correction). Since we do not employ readout-correction in the Bell test itself, we have to account for readout errors, see Eq. 1. If we were to include readout errors already in the experimental value of the concurrence C, the contribution of readout errors would be over-represented in Eq. 1. Therefore, it makes more sense to quote the concurrence of the Bell state in Fig. 2a, which lists the two quantities separately, in a readout-corrected manner.

3) In Abstract: "corresponding to a p-value of 10^{-108} "  "corresponding to a p-value no more than 10^{-108} "?

We revised the abstract to read *"we find an average S-value of 2.0747 +/- 0.0033, violating Bell's inequality with a p-value smaller than $p=10^{-108}$."*

4) In Appendix J: "as discussed in Appendix D"  "as discussed in Appendix E"? "Fig. J"  "Fig. 14"?

We have corrected the errors in the revised manuscript.

5) The possible values for the outcomes x, y in Eq. (K1) are not consistent with those in Tables III & IV.

In the revised version of the manuscript, we now define Eq. (K1) with the correct suffixes for the x - and y -values (+1 and -1 instead of 0 and 1).

6) Besides Refs. [75, 76], another experiment demonstrated device-independent randomness expansion. See the work [Phys. Rev. Lett. 126, 050503 (2021)] by Li et al.

While the data from Li et al. (PRL 2021) was present in the original Figure 16 (Figure S9 in the revised manuscript), the reference was missing from the caption of the figure. We have added the reference to the revised manuscript.

7) The work Liu et al. (2018) reported a p -value for rejecting local realism as small as $10^{-204792}$. Side comment: For the other demonstrations of device-independent randomness generation/expansion mentioned in Fig. 16, the p -values for rejection should be also extremely small, although they were not reported explicitly.

We have added a statement about this in the caption of Figure S9 in the revised manuscript: *"We note that many of the experiments realizing device-independent quantum information processing protocols should reach a low p -value, even though it is not always explicitly quoted. In particular, Ref. [Liu2018], marked with an arrow in the second plot, quotes a p -value of $p=10^{-204792}$."*

Overall, I consider the loophole-free Bell test reported in this work significant and impressive. I would therefore be happy to recommend acceptance, provided that the points mentioned above can be addressed or clarified.

We once more thank the referee for carefully reading our paper.

Referee #3 (Remarks to the Author):

Nature: Loophole-free Bell Inequality Violation with Superconducting Circuits

Summary and significance: The authors carry out an experiment in which they violate a CHSH Bell Inequality while addressing a combination of major loopholes, including the “locality”, “fair-sampling”, and “memory” loopholes. Their impressive setup consists of two superconducting qubit circuits, each coupled to its own control and readout electronics, and each coupled to opposite ends of a 30-m superconducting waveguide. All superconducting elements are contained within a single, large vacuum cryogenic system. The methods and supplemental materials offer a comprehensive analysis of the experimental design, construction, and data analysis, including a comparison to previous Bell experiments that have been labeled “loophole-free”. Although other setups in the past few years have been shown to close these loopholes simultaneously, this is the first experiment on the superconducting circuit platform to both address the locality loophole and close the fair-sampling loophole, and to my knowledge, it represents the largest physical separation demonstrated to date between two entangled superconducting circuits. Given the foundational significance of Bell-Inequality violations, and the promise that superconducting qubits have for realizing quantum computing technology, it is of great value to the community to see such an experiment on this platform.

The performance of this experiment is noteworthy compared to previous experiments. Photon-based implementations historically show low margins of violation with high data rates, which allow them to reach high statistical certainty of the violation; matter-based implementations historically show higher margins of violation but suffer from low data rates, which can make it difficult or at least time-consuming to reach high statistical certainty. This implementation yields a higher margin of violation than previous optical setups, and a higher data-rate than previous matter-based setups. It includes strong statistical claims generally in line with the strongest previous experiments.

Comments:

Defining Locality

The authors state in the first two sentences of the abstract (lines 18, 19) that, “Superposition, entanglement and non-locality constitute fundamental features of quantum physics” and “quantum physics does not follow the principle of locality”. Einstein’s locality, represented in special relativity theory, states that an object is directly influenced only by its immediate surroundings. Violations of Einstein-locality would enable signaling or faster-than-light communication of messages; this is decidedly not a feature of quantum theory. Quantum theory follows Einstein-locality.

The aspect of quantum theory EPR and Bell focused on in refs 1 and 2 is the situation that, in a bipartite quantum system, the joint outcome probabilities don’t factor. EPR and Bell investigate whether the addition of so-called “hidden variables” could enable each component system to be described independently. The worldview in which such hidden variables exist and are governed by Einstein-locality has been given many names even within the quantum community, for clarity for the moment, let’s call it Bell-locality. Bell showed that no model using (Einstein-)local hidden-variables

(LHVs) can reproduce all predictions of quantum mechanics, that is, quantum theory violates Bell-locality.

The distinction between Einstein-locality and Bell-locality is crucial, and is currently missing from this manuscript. There are three aspects of this discussion I'd like to see addressed.

1) Clearer definitions and language around "locality" throughout the main paper's text. Even if the manuscript were targeting only a small community of Bell-locality enthusiasts, it would benefit from clearer verbiage; given that the manuscript is targeting a broader audience, more specific language and definitions are critically needed. Throughout the manuscript, the authors use the general term "locality" to mean the more nuanced "Bell-locality" and "local models" to refer to LHV models. In fact, Einstein's locality predates Bell's, and a Wikipedia search for "principle of locality" describes Einstein-locality; an uninitiated reader could reasonably be confused or misled by the jargon, interpreting the manuscript to claim that quantum theory violates Einstein-locality—which is false. The manuscript would benefit from expressing clearly that the thing we actually test is a set of LHV models.

We thank the Referee for pointing out potential confusions. We fully agree with the Referee that Bell's notion of locality is distinct from the notion of locality whose violation would enable signaling or faster-than-light communication of messages. To avoid any confusion, we now use Bell's own terminology of 'local causality' to refer to the notion of 'locality' that he introduced [J. S. Bell, in *Speakable and Unsayable in Quantum Mechanics*, second edition, pp 232-248 (2004)]. We also renamed 'local model' into 'local hidden variable models'.

Moreover, we added a paragraph in the corresponding Appendix (lines 208-209 [Supplementary Information]) stating explicitly that violation of this notion of locality does not enable signaling. Furthermore, we added a subsection in Appendix X where we analyze the signaling character of our statistics. This analysis didn't reveal any significant trace of signaling in our data.

2) Cleaner statements about "realism" in the appendix. The end of Appendix A (line 657) includes a specific assertion that "no notion of realism appeared in our discussion so far, neither in the derivation of a Bell inequality, nor in the definition of the model. This shows that this concept is not necessary to interpret the result of a Bell test, as already noticed by Bell and others" In fact, the derivation of Bell's inequality in the original and CHSH forms include assumptions of Einstein-locality and determinism—the second component is left out of the discussion in Appendix A, but the omission from this manuscript does not remove it from relevance. In other works, some authors use the term "local realism" as a synonym for "Bell-locality"—meaning any theory of LHVs—as a way to differentiate from Einstein-locality. In fact, however we call it, a violation of Bell's inequality represents a violation of all covered LHV models, not the broader set of Einstein-local models. I'd encourage the authors to reconsider this paragraph, both in content and tone. At a minimum, the authors should please clarify that, even with their chosen notation/definition for "locality", a violation of Bell's inequality does not imply a violation of special relativity.

Following the Referee's suggestion, we added a paragraph to the corresponding Appendix where we clarify that the violation of a Bell inequality does not imply a violation of special relativity (lines 208-209 [Supplementary Information]).

We now also discuss the role of determinism in local hidden variable models (LHVMs), and show that determinism is not a necessary assumption in our derivation (lines 108-111 [Supplementary Information]). This is consistent with Bell's original derivation in that it is well known that a given conclusion can sometimes be obtained from different sets of assumptions. In the case of Bell's theorem, this phenomenon has been precisely documented in [H. M. Wiseman, J. of Phys. A: Math. and Theor. 47, 424001 (2014)], which we cite.

3) Reference 37 (line 2015) needs some attention. It references the entire "Speakable and Unspeakable in Quantum Mechanics" book, which is a compendium of Bell's papers on the topic. I recommend two adjustments:

a) The book is a compendium of 24 papers. The citation should include a more specific reference—if not the page number then at least the paper title.

b) Aspect, who wrote the forward, is listed as a co-author. However, the book is entirely Bell's work. Unless the reference is intended to cite something in the forward, Bell should be the author.

We thank the referee for pointing this out. In the revised manuscript we have corrected the author list of Ref. 3, and we explicitly refer to Chapter 24 (La nouvelle cuisine, pages 232-248) in the book.

Inconsistent definition of "measurement independence"

The term "measurement" is an ambiguous word, especially in quantum contexts, as it may not distinguish between measurement inputs, outputs, time, processes, parties, etc. Even within this manuscript, the term "measurement independence" is used inconsistently. I invite the authors to both remove the inconsistency and possibly replace the term "measurement independence" with something more specific.

Line 57: "measurement independence, the idea that the choice of the two possible measurements is statistically independent from the qubits and the measurement devices"

What does it mean for the choice to be independent from the qubits and measurement devices? Independent from possible LHVs located in the qubits? From measurement outcomes? Something else?

In this case, between lines 53 and 60 I think the authors are trying to capture the assumptions of

- 1) independence between each setting choice and the distant measurement outcome
- 2) independence between each setting choice and any hypothetical hidden variable and the text could be refined to clarify this.

Line 573: "measurement independence condition $P(\lambda|ab) = P(\lambda)$ "

Here, the authors explicitly define the condition as independence between each setting choice and any hypothetical hidden variable (#2 above).

We thank the Referee for pointing out that the definition of measurement independence provided in the introduction was confusing. We modified the text to read (lines 59-61 in the revised manuscript [Main text])

“measurement independence, the idea that the choice between the two possible measurements is statistically independent from any hidden variables”

which makes it clear that the independence is between the measurement choice and the hidden variable. We believe that this formulation corrects any perceived inconsistency. In particular, it leaves no doubt on the part of the measurement involved, namely its choice (which the Referee also calls the input, the setting choice and simply the setting). This corresponds precisely to point (2) expressed by the Referee, and is its only intended meaning in the manuscript. Measurement independence is never intended to be used as in case (1).

The ‘measurement independence’ terminology has been used by a wide array of works, notably in [M. Hall, Phys. Rev. Lett. 105, 250404 (2010); J. Barrett and N. Gisin, Phys. Rev. Lett. 106, 100406 (2010)], who first proposed ways of quantifying ‘measurement dependence’, thus raising awareness on what had often been considered a purely philosophical question. We believe that the term of ‘measurement independence’ is the most precise terminology to describe this condition of statistical independence.

Missing/incomplete discussion of setting-LHV independence.

Two kinds of setting independence are assumed in any physical implementation leveraging Bell’s inequality to test a local-hidden-variable (LHV) worldview.

- 1) setting-outcome independence: Independence between Alice’s (Bob’s) measurement setting choice and Bob’s (Alice’s) measurement outcome. Synonymous with “locality loophole”.
- 2) setting-HV independence: Independence between each measurement setting choice and any hidden variables (HV).

To establish independence of the settings—whether from outcomes (1 above) or from hidden variables (2 above)—the experimentalists employ two strategies:

StrategyA: enforce space-like separation between events that should be independent

StrategyB: appeal to a physical model about the mechanics of the RNG

Of course, there will be LHV models that cannot be excluded in any feasible experimental construction. Among others, superdeterminist models could be responsible for violations of both setting-outcome independence and setting-HV independence, and cannot be excluded through any physical strategies. In lines 636-640, the authors mention specifically that superdeterminist models cannot be excluded, however, there is no mention of other models that can be excluded, such as some where a causal relationship exists between LHVs and setting choices. The goal of a “loophole-free” experiment is to exclude as many LHV models as possible, and represent as thoroughly as possible the categories of LHV models that have been excluded.

We thank the Referee for these comments and we agree that a “loophole-free” experiment should exclude as many LHVMs as possible, i.e. exclude LHVMs with as few assumptions as possible. We note that this matches the widely-used criteria from [J-A. Larsson, J. Phys. A: Math. Theor. 47, 424003 (2014)], which states that:

“In general, assumptions other than local realism should be absent from a loophole-free experiment”.

We also agree with the Referee that we did not discuss the situation where a causal relation exists between the hidden variable λ and the settings choices a and b . In such situations, the relation $p(\lambda|a,b)=p(\lambda)$ (or equivalently $p(a,b|\lambda)=p(a,b)$, we will use them interchangeably), doesn't hold and the standard derivation of Bell's inequalities doesn't apply anymore. However, it was shown recently that local hidden variable models can sometimes still be ruled out under the weaker condition $p(a,b|\lambda) \geq I$, as long as $I > 0$ [G. Pütz, D. Rosset, T. J. Barnea, Y.-C. Liang, and N. Gisin, Phys. Rev. Lett. 113, 190402 (2014)]. In this case, a Bell inequality, generally distinct from CHSH, can tell apart local hidden variable models. We thank the Referee for pointing out this important extension of the usual Bell analysis. It is now duly mentioned in Appendix A.

As far as we know, all other experimental situations where a causal relation exists between the hidden variables and the setting choices are unable to rule out LHVMs without additional assumption. Such situations thus diverge from the purpose of “loophole-free” experiments cited above.

Additionally, let us mention that in our experiment, the entanglement is produced efficiently and deterministically, but not rapidly compared to the light-travel time between the measurement stations. For this reason, space-time considerations do not play a role in determining setting-HV independence. Rather, this is determined by the ability of HVs to influence the RNGs (see also discussion below). The RNGs operate in a similar manner to those used in Hensen et al. (2015), Giustina et al. (2015) and Shalm et al. (2015), and were designed to exclude specific classes of LHVMs.

As described in Appendix II and also in Abellan et al., PRL 115 (2015), the RNGs output is almost entirely randomized by the process of laser phase diffusion. The residual influence of other effects is quantified by the parameter ϵ , which is upper-bounded under mild assumptions by $5.24e-6$. The p-value is computed assuming the LHVM takes maximum advantage of this (small) degree of possible influence. Consequently, we can say that the experiment excludes (subject to mild assumptions) any LHVM that does not influence laser phase diffusion. Beyond that, we note that laser phase diffusion is caused by multiple factors, including spontaneous emission, thermal fluctuations of laser cavity parameters, and carrier density fluctuations. Following diffusion, the resulting phase is random if any one of these influences is random. We can thus say (under mild assumptions) that the experiment excludes any LHVM which do not influence all of these factors.

To achieve setting-outcome independence: In appendix A of this work, the authors carefully detail how they rely on space-like separation (StrategyA) to convert {locality condition} into {principle of locality, space-like separation} to approach setting-outcome independence using space-like separation.

To achieve setting-HV independence: There is no explicit discussion of how to approach setting-HV independence. Given the inconsistent definition of “measurement independence” (see above), it's possible the authors mean lines 671-680 of appendix B as an appeal to StrategyB to achieve setting-HV independence. There is no mention of space-like separation.

As discussed above, we believe that any ambiguity about the meaning of the measurement-independence condition has now been removed. Measurement independence corresponds to the

condition $P(a,b|\lambda)=P(a,b)$, which the Referee calls ‘settings-HV independence’. In turn, this mathematical relation matches exactly the kind of condition that is provided by a physical model of a RNG device, hence StrategyB addresses it fully. In other words, settings-HV independence can be based directly on the physical modeling of the RNG devices, as we do. This is mentioned in the main text of the revised manuscript as (lines 123-125 [Main text]):

“To support the assumption of measurement independence, local basis choices are realized using random number generators.”

Note that settings-HV independence was also justified by StrategyB only in [W. Rosenfeld et al., Phys. Rev. Lett. 119, 010402 (2017)]. Indeed, very much like in our case, this experiment performed a Bell test in its most natural fashion, where an outcome is recorded every time a setting is chosen. When this is the case, the condition of settings-HV independence identifies with the characterization of RNGs, and is thus fully addressed by StrategyB.

In contrast, in the experiment of [Hensen et al. (2015)] outcomes were not produced by the nodes every time the RNGs were used. Indeed, the outcomes were only kept when the central station detected a photon from both nodes. This leads to a situation akin to the detection loophole, which happens when outcomes are discarded in case of no-detection. Namely, here, this opened the possibility for the central station to select hidden variables according to the settings chosen by the RNGs, i.e. to choose a $P(\lambda|ab)$ different from $P(\lambda)$ and thus break settings-HV independence. The way this was avoided in the experiment was by making sure the heralding event was space-like separated from the RNG events, i.e. by using a combination of StrategyB with StrategyA as suggested by the Referee. We emphasize that this was necessary due to discarding rounds where settings have been chosen, which is well known to open loopholes. In our experiment, this situation does not apply as we do not discard any rounds and do not employ a heralding scheme.

Note that another case has also been considered in the literature where the justification of settings-HV independence involved StrategyA only. This was the case in the photonic experiments of Giustina et al. and Shalm et al., where space-like separation was enforced between the photon pair creation event and the measurement choice events. Space-like separation ensures that HVs created at the space-time point associated with the photon pair creation are not able to influence the measurement choices. If the HVs are, in addition, independent of prior events, it follows that the HVs and the measurement choices are independent, achieving 2).

If, on the contrary, the HVs are statistically related to some prior event, either because they are related to that event at their “birth,” or are influenced by it later, they are correlated with prior events that could sub-luminally influence the measurement choices. This breaks HV-setting independence. It is clear then that there is a class of LHVMs for which StrategyA achieves 2). But it is hard to argue that it is an important class: First, the entanglement generation event is important in quantum physics but has no special relevance in LHVMs, because entanglement is a quantum, not HV concept. Second, since we are talking about a mysterious microscopic event (either a measurement or production of photon pairs) it is difficult to argue that the HVs produced through this process should be independent of prior events. Finally, there is no particular reason to think that HVs, even if “born free,” i.e., independent of the past, will remain independent of prior variables, because they could be influenced after they are created.

Furthermore, in our experiment, the “entanglement generation event” is not well-approximated by a point in space-time, but is in fact a relatively long process. Consequently, it is not possible to achieve space-like separation between this process and the measurement events, and StrategyB is the only available strategy.

We thank the referee for this discussion, and we have modified the text in Methods to clarify these points, see lines 593-641 in the revised manuscript [Main text].

These two approaches are inconsistent. If StrategyB is sufficient for claims of independence, then space-like separation would not be necessary to address the locality loophole. Likewise, if space-like separation can help address the locality loophole—which is generally held to be the case, and which the authors argue in Appendix A—then it can also play a role in building setting-HV independence.

We agree that space-like separation can sometimes play a role in guaranteeing settings-HV independence, see discussion above. In our case however, similarly to other experiments mentioned above, StrategyB is sufficient.

This manuscript would benefit significantly from:

- A) A direct discussion about setting-HV independence and the role it plays in this experiment.
- B) An analysis of setting-HV independence from a space-time perspective, indicating which LHV models could be restricted based on the space-time configuration used in this experiment.
- C) An extension of the argument in lines 636-640 to address also the non-superdeterminist models where setting-LHV independence can(/not) be established.

Furthermore, with the omission of discussion about setting-HV independence, this experiment glosses over a loophole that has been addressed in other previous experiments. Especially if this manuscript claims to be “loophole-free”, a rigorous discussion of setting-HV independence is in order.

We believe that by the changes made in the revised manuscript, we have satisfied the above requests by the Referee.

Once again, we thank the Referee for very insightful comments and criticisms. Indeed, the manuscript has, we believe, benefitted significantly from the changes made in response to these comments.

Definition and claims of “loophole-free”:

Consider using “loophole-free” with a little more care:

- Line 66: “loophole-free Bell inequality violations, which do not rely on additional assumptions, were demonstrated in 2015 and the following years...” Ultimately, no test of Bell’s inequality can be truly free of additional assumptions—there’s no such thing as a truly “loophole-free” experiment. We can exclude different and ever larger sets of possible LHV models, but there will always be residual assumptions. Thus the statement that any violations “do not rely on additional assumptions” is too strong.
- This experiment is clearly a tour de force, and that speaks for itself. I would encourage using the word “loophole-free” sparingly—it appears 14 times in the main text and another 11 times in the

supplement. When it's possible, an accurate description of the experiment is more compelling than a shiny buzzword that's inevitably a stretch. It is in the interest of the authors and the community to communicate about these experiments with meticulousness and care.

We thank the Referee for these two comments. We consider the word “loophole-free” as a qualifier for a specific type of experiment, and we agree with the referee that at times this qualifier is not used very carefully. In most literature, it has become a common term in the community for Bell tests that simultaneously close the detection and the locality loophole, and address measurement independence (sometimes referred to as closing the freedom of choice loophole). This is also the spirit of how the term was defined in [J-A. Larsson, J. Phys. A 47 424003 (2014)], which we repeatedly cite in the publication. We therefore prefer to keep the terminology, but – as the Referee suggests – we use the term more sparingly in the revised version. Furthermore, as suggested, in line 68 in the submitted manuscript [Main text] we now write “(...) until loophole-free Bell inequality violations, which close all major loopholes simultaneously, were demonstrated (...)”.

Measurement, readout and reset:

The section on readout is clear and pretty thorough. Still, it would benefit from a statement about the reset fidelity, including the time it takes to achieve. I'd assume the authors wait between trials for long enough for the readout resonator to reset, a statement of the time and performance would be valuable.

In the revised manuscript, we have added the following discussion to the corresponding appendix VII (qubit readout, lines 879-894 [Supplementary Information]):

“We note that for all experiments discussed in this manuscript qubit reset was implemented at the beginning of each individual pulse sequence, including for characterization of the readout protocol. The reset scheme we employ is described in detail in Ref. [Magnard et al., PRL 121, 060502 (2018)], based on driving the same $|f0\rangle \leftrightarrow |g1\rangle$ transition which we also use for the generation of entanglement (Supplementary Section VI). In this continuously driven reset-scheme, population is transferred from the first two excited states of the qubit to excitations in the transfer resonator. While we drive the reset for $1 \mu s$, most of the population of the first two excited states of the qubit is transferred to the resonator within the first 100 ns of the protocol. The remaining excited state population of the qubit after applying the reset scheme is below the detection threshold of our measurement scheme, which is below 1%.”

The authors point out the presence of a TLS near one of their qubits, which contributes to a lowered readout fidelity. I'm curious if the T1 of this TLS is known. Given the statistical analysis the authors use to close the memory loophole, any possible memory between trials—such as a possible memory mediated by the TLS—could not have an effect on the conclusion, so strictly speaking there's no need to include this analysis. I'm just curious. :-)

We have not explicitly measured the lifetime of the defect. We agree with the Referee that any possible memory between trials mediated by the TLS could not have an effect on the conclusion as our statistical analysis is robust to memory effects.

Line 1785: What is meant by “classical noise”?

We have clarified this point in the revised manuscript by changing the phrase as follows:

“We amplify the signal a second time using a high-electron-mobility transistor (HEMT) amplifier, located at the 4K stage of the dilution refrigerator. This is the first amplification (about 40 dB) that is not quantum-limited, adding noise photons that are uncorrelated with the signal [C. Eichler, PhD thesis, ETH Zurich (2013)].”

Figures:

- Figure 1 doesn't do much for me—it's not clear what this figure accomplishes that is not already represented in figure 3 or figure 4. On a first read, I found the figure distracting because I was looking for more information. Consider sharpening the focus or removing the figure.

Figure 1 is intended to help a non-expert reader to understand the concept of a Bell test experiment. In contrast to Figures 3 and 4, it does not contain technical details about the experimental setup, which could potentially overwhelm the broad audience we are targeting from understanding the core concept of the experiment in the beginning of the paper. We therefore believe Figure 1 is an asset to the paper and we prefer to keep it in the publication.

- Figure 4: consider adding uncertainties for the time values at the far-right of the figure.

The main goal of Figure 4 is to visualize the main segments of the protocol in a compact manner. We are of the opinion that adding uncertainties will make it harder to understand the protocol, as it distracts from the main focus of the figure. Furthermore, we do not have precise values and uncertainties for all the displayed segments (e.g. the signal propagation delay in the cables). This is because for the measurement of the total protocol duration we combined measurements of the duration of a different set of protocol sub-segments, which were more easily accessible experimentally (see Appendix XI and Figure S4 in the revised manuscript).

- Figure 7 is exhaustive. Can it be simplified? For example, it looks like the Alice and Bob setups are almost identical, and there are so many elements that it's difficult to identify similarities and differences. Consider drawing just one analysis setup, and highlight (with description in text and/or figure caption) any element that exists only within Alice or Bob.

We agree with the referee that Figure S5 contains a lot of information and we thank for the suggestion to simplify the figure by exploiting the symmetry in the electronic setup.

In the revised Figure S5 we have made the elements on Bob's setup translucent that are equivalent to Alice's node. We have also commented on that in the caption of Figure S5.

Nit-pick questions and suggestions:

- Line 123: “occurring at best at the speed of light” → “occurring at most at the speed of light” or “occurring at the speed of light or slower”

We have changed the wording accordingly in the revised manuscript: “(...) occurring at most at the speed of light c (...)”

- The text refers a few times to “node C” (e.g. line 903, 908-9, 932, and possibly others), but I don’t see “node C” labeled in any diagram.

In the first paragraph of Appendix V.B we have now clarified what nodes A, B and C stand for.

- The word “qutrit” appears without introduction several times in Appendix H and Appendix I. The manuscript would benefit from a clearer definition of what elements are meant with “qutrit”, and how those are distinguished from what is meant with “qubit”. For example, line 1415 includes “emitter qubit in the $1/\sqrt{2} (|g\rangle + |f\rangle)$ state”, but elsewhere the qubit’s states are given as $|g\rangle$ and $|e\rangle$. So should line 1415 reference the qubit, or qutrit? Please review both appendices with this distinction in mind.

In the revised manuscript we have added (on the request of Referee 2) a Methods section on superconducting qubits (lines 499-525 [Main text]), which contains an explanation of when we refer to the quantum system as a qutrit. Furthermore, as suggested by the referee, we have revised both appendices H and I (new labelling: VI and VII) for a more consistent use of the terms qubit and qutrit.

- Line 1693-1697: “For experiments using optical photons... the CHSH violation typically is low...” Note that experiments using optical photons closing the fair-sampling loophole typically use the CH-inequality, since the CHSH-inequality assumes fair-sampling while the CH-inequality can be violated only by an experiment showing adequately high collection efficiency.

We understand that the reader may be confused by the two different types of S-values. For this reason we have added a clarifying statement in the text of Appendix VIII (which was previously partially addressed in the caption of Figure S9):

*“(Some publications quoted the CH-Eberhard [Eberhard1993] instead of the CHSH S-value. For a more straightforward comparison we converted the respective numbers to the CHSH S-value using $S_{CHSH}=4*S_{CH}+2$ both in the text and in Figure S9.)”*

- Line 1810-1811: “The ultimate length limit of the current setup is given by the length of the laboratory...” This sounds like a current limit, but not an in-principle limit. Since this discussion is focused on how to increase the margins, it would be interesting to hear what in-principle limits exist for this kind of setup.

In response to the question we have added the following statement in the revised manuscript:

“In principle, and supported by heat-flow simulations based on the measured thermal properties of the system, the presented cryogenic-link architecture could be extended to distances of several 100 m when placing a 4K cooling unit every 15 m and a dilution refrigerator every 100 m. Increasing the length of the cryogenic quantum channel to km-scale distances could be achieved with optimizations in the design of the system and the selection of materials. Operation of the waveguide at elevated temperatures, as suggested in [Z/L. Xiang et al., Physical Review X 7, 011035 (2017)] may also increase the maximum separation that can be achieved between link nodes.”

We note that we plan to discuss design aspects of the cryogenic system in greater detail in a separate manuscript.

We again thank the referee for the careful reading of our paper, and we believe that the suggestions made by the referee have significantly improved the manuscript.

Reviewer Reports on the First Revision:

Referees' comments:

Referee #1 (Remarks to the Author):

I thank the authors for comprehensively addressing the remarks from my first report. All points have been addressed satisfactorily. I recommend the revised manuscript for publication in Nature.

A few very minor issues arose in the revision process, which should be addressed prior to publication. The authors can address these points without my needing to see the manuscript again.

- In the new Methods section "Superconducting Qubits," the word "with" is misspelled at line 517. (That whole sentence is a little complicated with many clauses and could perhaps merit a small re-write.)

- In the new Methods section "The p-value as a Statistical Metric," I find the following sentence to be mis-stated (line 658): "It is now an established practice [6-10] to calculate the p-value of the result, which does not rely on any of the aforementioned assumptions." This seems to imply there is some canonical "p-value method" which is robust to the mentioned issues (e.g., memory loophole). However, one could easily use some statistical method invoking i.i.d. assumptions, opening the memory loophole, and calculate a p-value - the p-value is just a common statistical metric that can be computed for any statistical hypothesis test. Furthermore, there are also more than one memory robust statistical analysis methods for analyzing Bell data that result in p-values. Really what the authors mean is that they "calculate the p-value of the result [according to a method that does not make] any of the aforementioned assumptions." The specific method of the authors is a binomial distribution approach rigorously proven to be memory robust in SM Ref. [1], whose basic idea can also be seen in [Phys. Rev. A 66, 042111 (2002)], and the method is adapted to encompass small deviations from perfect measurement settings independence following the treatment in the supplementary material of Ref. [6].

- At SM line 937, "This combination allows us to reach an unequaled statistical significance (p-value) of $p = 10^{-108}$." This read to me as possibly overstated the first time through. With the figure $10^{-204792}$ now included in the caption of Fig. S9 (this is new to the revised manuscript), the adjective "unequaled" seems to need replacement. It is true that 10^{-108} is still a strong figure relative to most other loophole-free p-values.

- Line SM 1381: Now that the revised manuscript has made it clearer that r_{start} and r_{end} are spatial positions (not space time events) I rescind one of my original critiques and believe that $||r_{\text{start}} - r_{\text{end}}||$ as originally notated with double-lines (norm notation) is mathematically appropriate and can be used.

Referee #2 (Remarks to the Author):

First of all, I would like to thank the authors for all the clarifications and explanations made in their revised manuscript and their resubmission response. The authors have addressed most of the points of criticism and comments raised in my first report. I also think the manuscript becomes more accessible to general readers. At the same time, I observe one point that could be clarified further and catch a few minor issues in the revised manuscript. Before recommending the acceptance of this impressive and significant work, I would like to ask the authors to check the following points:

1) About the interpretation of a Bell-inequality violation: The use of hidden variables to describe experimental phenomena is a pre-condition for us to interpret that a Bell-inequality violation indicates that either the locality or measurement independence cannot be satisfied by experimental results. In my opinion, the use of hidden variables corresponds to an additional assumption underlying the null hypothesis falsified by a loophole-free Bell test. In my previous report, I thought this additional assumption is captured by the notion of realism. I accept that the authors may not agree with me. However, I recommend the authors emphasize the use/role of hidden variables in their derivation of Bell inequalities and clarify the definition of realism if the authors don't think their derivation relies on the notion of realism.

I also would like to add the following: The conclusion of a loophole-free Bell test could be that hidden variables cannot restore locality (a conclusion preferred by John S. Bell and others). In other words, a hidden-variable description of experimental phenomena must forgo the principle of local causality. However, it doesn't imply that there is no quantum-mechanical description compatible with locality.

If the authors agree with me, I hope the authors can clarify the relation summarized in Eq. (11) of the SI. In particular, I hope the authors can emphasize that a hidden-variable description is as important as the other two assumptions (locality and measurement independence) in order to derive a Bell inequality. A similar clarification can be made for the relation summarized in Eq. (12) of the SI.

2) To obtain Eq. (5) in Section I of the SI, it is mentioned in lines 89-90 that Bayes' rule is used. Instead of Bayes' rule, I think the chain rule in probability theory should be used.

3) In lines 86, 87, 88, 101, 102, and 108 of the SI, the distribution of the hidden variable λ is denoted by the little-case letter p , which is not consistent with the notation introduced in line 77.

4) The p -value is defined in the first sentence (lines 1218-1220) of Section X.B in the SI. I think this definition could be more precise. There are many different local hidden variable models, and the probabilities according to different models of producing statistics at least as extreme as the experimental one can be different. Given these, one can figure out the best local hidden variable model according to which the probability of producing a statistic at least as extreme as the experimental one is the highest. Such the highest probability is the p -value for testing local hidden variable models.

5) About the success probability p_{win} in Eq. (19) of Section X.B in the SI, it is not clear from that equation whether the success probability is according to a local hidden variable model or according to a quantum strategy. I suggest the authors clearly define the success probabilities according to different strategies. For example, the authors could introduce two variables $p_{\text{win}}^{\text{LHV}}$ and $p_{\text{win}}^{\text{QM}}$ in the first paragraph of Section X.B. Then it would become clear which success probability should be referred to in the equations below that paragraph.

6) In lines 1326-1327 of the newly added Section X.C, it is mentioned that a_i (b_i) and x_i (y_i) are the outcome and setting of Alice (Bob) observed in round i of the experiment. The notation here is not consistent with that used elsewhere in the manuscript: In this work, a and b are used to denote the settings of Alice and Bob, and x and y are used to denote their outcomes. I also find that the term "round" has been used several times in Section X.C, while elsewhere in the manuscript the term "trial" has been used.

Referee #3 (Remarks to the Author):

Nature: Loophole-free Bell Inequality Violation with Superconducting Circuits

The authors have implemented significant updates to the manuscript in response to the first round of revision requests. I have a few additional comments, some about elements in the original manuscript that have become more clear with the revision, and some about the revised sections. Some of these may seem like language nits, however for this topic in particular it is valuable to use language with careful intention.

Main text

Line 65: "early experiments relied on additional assumptions, creating loopholes in Bell's argument."
>> No, Bell's argument is sound. It is the mapping of a gedankenexperiment into the lab that introduces the assumptions and thus the loopholes.

Line 71: "As quantum technology matured..." this suggests that quantum technology has matured; arguably it's still pretty nascent. Consider alternate verbiage, e.g. "In the development of quantum information science, ..."

Line 80: "(the choice of measurement)" The word "measurement" is especially ambiguous when it gets close to quantum mechanics—is this the choice of what to measure, how to measure, when to measure, whether to measure...? Consider being more explicit: "(the choice of measurement basis)".

Line 108-111: "While these experiments all relied on additional assumptions, in this work, we set out to demonstrate a loophole-free violation of Bell's inequality using superconducting circuits."

>> In my eyes, stating which loopholes an experiment claims to close is important to the overall claim of the paper. I could not find a place in the manuscript where it is articulated which loopholes will be addressed. No experiment is truly "loophole-free", and the term is vague enough that a case-specific description is warranted—this seems like a great spot to add it. (Yes, there is a citation of

(Larsson, 2014), however this reference includes mention of a number of loopholes including some that aren't actually relevant to this setup. Regardless of the citation, any strong manuscript should describe clearly at least once which loopholes it claims to close.)

Line 114: The "locality loophole" is introduced without explanation in the main text. Its definition is central to the manuscript's claims. Consider inserting a reference to the Supplementary information where the loophole definition is more clearly explained. (This is also relevant in line 368.)

Line 119: To appeal to a broad audience, consider defining "concurrency"

Line 132-4: Consider the following adjustment for clarity/simplicity: "the corresponding measurement outcomes by the party at one site are unknown to the party at the other site"

Line 135: "The locality condition (Supplementary Section I) is also commonly considered as a fundamental ingredient of a Bell test rather than a loophole [3, 19]."

>> My impression: this sentence almost seems to undermine the entire section, it's not clear to me that it adds strength or clarity to the claims. Consider removing it. If you must keep it, consider relocating it to the Supplement.

Line 145: Like the comment in Line 114, the "detection loophole" is not defined; unlike the comment in 114 I can't find a definition in the supplement. I suspect the authors actually want to refer to the "fair-sampling" loophole; the term "detection loophole" is rather informal, and originated in photon experiments where a fair-sampling assumption may come as a result of low photon detection rates. Consider updating and defining (somewhere) the loophole term.

Line 156-7: "If the properties of the two entangled qubits were described by a local hidden variable model"

>> For strength of argument, consider "If the properties of the system were described..." The goal of a Bell experiment is to test a system's ability to be described by hypothetical LHVs; we need not assume that said system is quantum in order for the test to be valid.

Line 162-163: "realizing a Bell test closing all major loopholes" >> this would be another great place to reference the specific list of loopholes the experiment aims to close.

Line 184: nit. "which" → "that"

Line 187: Please remove "to assert non-locality". The experiment doesn't "assert non-locality", rather it leverages space-like separation to address the locality loophole.

Line 212: For clarity, consider "due to" → "by leveraging". (I had to read this sentence a few times before I understood it.)

Line 223: "major cryogenic system" I'm puzzled by the word "major"; Is this well-defined jargon? Maybe "large-scale"?

Lines 267-269: “We consider the start event of each trial of a Bell test as being marked in space and time by the creation of a random number in each RNG.”

>> Each trial should have a single start event; here two are mentioned (RNGA and RNGB).

Lines 273-275: “a and b become available as voltage pulses at the output of the RNGs 17.10 ± 0.14 ns after this event”

>> Perhaps connected to the “multiple start events” confusion from lines 267-269, I got a little confused by this sentence. Consider making this sentence a single-RNG sentence, rather than a general statement about the overall experiment, e.g. “Each setting output from a RNG (a and b respectively) becomes available as a voltage pulse at the RNG’s output 17.10 ± 0.14 ns after the random number’s creation.”

Lines 280-287: I struggled to match the text here with the corresponding blocks in figure 4, in part because figure 4 contains five separate blocks that could be interpreted as close to “dark green”, and in part because the text in these lines doesn’t match word-for-word with the figure labels. For me, establishing matching words between the manuscript text and the figure would be enough.

Line 299: For clarity, consider “As for” → “As done with”. (I had to read this sentence a few times before I understood it.)

Line 301-2: For clarity, consider “In this way we minimize the propagation delay... to 14ns...” → “In this way we minimize to 14 ns the propagation delay...”. (I had to read this sentence a few times before I understood it.)

Lines 417-425: This seems to mostly repeat the prior paragraph, feeling a little rambling. Consider distilling the new material into a single DI-focused sentence and cutting the rest.

Lines 430-433: “Interconnected cryogenic systems may indicate a pathway towards realizing larger scale quantum computing systems using quantum microwave local area networks.”

>> This is very surprising to me—naively I’d expect this type of setup to be prohibitively challenging to deploy at scale! Perhaps the authors have learned, through building this experiment, about some practical challenges one would face to deployment at scale, or what makes this easier than expected. If so, that would be very interesting to hear.

Line 517: “with” ?

Line 518, Supplement section IV “Quantum Devices”: Consider adding just one more explanatory sentence in at least one of these locations to explain why both “qubit” and “qutrit” appear in the manuscript and how a reader should interpret when they see one vs the other—otherwise the switch (paper mostly uses qubit, supplement mostly uses qutrit) is a little disorienting.

Lines 609-612, 613-641: “This consideration has no relevance to our situation, because we use a deterministic entanglement generation protocol which is independent of the outcome of any measurement.”

>> The section in 613-641 is long, verbose, and slightly dismissive; another (simpler) way to

articulate this is that for a given assumption about setting choice origin (which is made already in order to address the locality loophole) there are certain possible origins of λ that can be excluded by the experiment due to space-time layout, and other possible origins of λ that cannot be excluded through space-time layout. This statement affects the present experiment just like any other. Thus to suggest that “This consideration has no relevance to our situation” seems to miss the point. If the goal is to address ref#2’s request, consider an actual space-time comparison; at least please adopt a less dismissive tone.

Supplement

Lines 123-124: “For instance, in a space-time configuration allowing for communication from A to B,”
>> Since A and B define spatial locations only, any configuration allows for such communication.

Lines 133-135: “On the other hand, a theory that is not locally causal will be referred to as non-local”.

>> Per previous discussion of the term “non-local”, this could be misleading. The definition also is not used elsewhere. Please remove this sentence.

Lines 136-138: “In order to test locally causal theories, we are interested in configurations where their statistics admit a local hidden variable model.”

>> I don’t understand what this sentence is supposed to communicate, in particular the phrase “their statistics”.

Lines 162-165: “In particular, this conclusion requires an appropriate definition of the events at which the inputs a and b are created and those at which the outcomes x and y are fixed.”

>> Consider a cleaner phrasing: “In particular, this conclusion relies on an assumption about the space-time points at which a and b are created and at which x and y are fixed.”

Lines 165-168: “Still, this shows that under the principle of local causality, the locality condition (2) reduces to an assumption on the time and location at which events happen.”

>> This phrasing is problematic (esp “reduces to”), and I don’t see that this sentence adds anything. Consider removing it.

Lines 185-186: “The amount of measurement independence needed to conclude can however...”

>> Needed to conclude what?

Lines 200-208: The authors state that “realism” is not needed and therefore their conclusion is clearer. I actually found the discussion leading up to this statement was less clear than other parts of the manuscript, so would shy away from emphasizing clarity here. I don’t understand what the authors refer to under “realism”, and thus I don’t understand what exactly they claim to avoid. For example, some might define realism as “LHVs exist”--which is an inevitable component of discussing hypothetical LHV models. I’d encourage the authors to clarify what they mean by realism, or better yet, to increase clarity, remove the section entirely. In my eyes, this section weakens, not strengthens, the manuscript.

Line 213: “fact” → “assertion”

Line 409: remove “non-local”

Line 544: “to at” >> pick one

Fig S5: The updated figure is so much easier to understand. Thank you for the modifications, I find them very helpful.

Author Rebuttals to First Revision:

Reply to the Referees

19th of February, 2023

Note from the authors:

In the following document we show the referee's comments in black and display our responses in blue.

Referee #1 (Remarks to the Author):

I thank the authors for comprehensively addressing the remarks from my first report. All points have been addressed satisfactorily. I recommend the revised manuscript for publication in Nature.

A few very minor issues arose in the revision process, which should be addressed prior to publication. The authors can address these points without my needing to see the manuscript again.

- In the new Methods section "Superconducting Qubits," the word "with" is misspelled at line 517. (That whole sentence is a little complicated with many clauses and could perhaps merit a small re-write.)

We have corrected this typo in the resubmitted manuscript.

- In the new Methods section "The p-value as a Statistical Metric," I find the following sentence to be mis-stated (line 658): "It is now an established practice [6-10] to calculate the p-value of the result, which does not rely on any of the aforementioned assumptions." This seems to imply there is some canonical "p-value method" which is robust to the mentioned issues (e.g., memory loophole). However, one could easily use some statistical method invoking i.i.d. assumptions, opening the memory loophole, and calculate a p-value - the p-value is just a common statistical metric that can be computed for any statistical hypothesis test. Furthermore, there are also more than one memory robust statistical analysis methods for analyzing Bell data that result in p-values. Really what the authors mean is that they "calculate the p-value of the result [according to a method that does not make] any of the aforementioned assumptions." The specific method of the authors is a binomial distribution approach rigorously proven to be memory robust in SM Ref. [1], whose basic idea can also be seen in [Phys. Rev. A 66, 042111 (2002)], and the method is adapted to encompass small deviations from perfect measurement settings independence following the treatment in the supplementary material of Ref. [6].

In the Methods section “The p-value as a Statistical Metric” we have now followed the suggestions of Referee 1 and write: “These two limitations can be addressed by the statistical analysis of the result through the calculation of a p-value according to a method which does not rely on any of the aforementioned assumptions. Therefore, the calculation of p-values in the context of Bell tests is now an established practice [6-10].”

- At SM line 937, "This combination allows us to reach an unequaled statistical significance (p-value) of $p = 10^{-108}$." This read to me as possibly overstated the first time through. With the figure $10^{-204792}$ now included in the caption of Fig. S9 (this is new to the revised manuscript), the adjective "unequaled" seems to need replacement. It is true that 10^{-108} is still a strong figure relative to most other loophole-free p-values.

We have replaced “unequaled” with “strong” in this sentence.

- Line SM 1381: Now that the revised manuscript has made it clearer that r_{start} and r_{end} are spatial positions (not space time events) I rescind one of my original critiques and believe that $|| r_{\text{start}} - r_{\text{end}} ||$ as originally notated with double-lines (norm notation) is mathematically appropriate and can be used.

We have also verified this aspect once more and agree that the norm notation is indeed the proper mathematical formulation. We have adapted this in the resubmitted manuscript.

Referee #2 (Remarks to the Author):

First of all, I would like to thank the authors for all the clarifications and explanations made in their revised manuscript and their resubmission response. The authors have addressed most of the points of criticism and comments raised in my first report. I also think the manuscript becomes more accessible to general readers. At the same time, I observe one point that could be clarified further and catch a few minor issues in the revised manuscript. Before recommending the acceptance of this impressive and significant work, I would like to ask the authors to check the following points:

1) About the interpretation of a Bell-inequality violation: The use of hidden variables to describe experimental phenomena is a pre-condition for us to interpret that a Bell-inequality violation indicates that either the locality or measurement independence cannot be satisfied by experimental results. In my opinion, the use of hidden variables corresponds to an additional assumption underlying the null hypothesis falsified by a loophole-free Bell test. In my previous report, I thought this additional assumption is captured by the notion of realism. I accept that the authors may not agree with me. However, I recommend the authors emphasize the use/role of hidden variables in their derivation of Bell inequalities and clarify the definition of realism if the authors don't think their derivation relies on the notion of realism.

I also would like to add the following: The conclusion of a loophole-free Bell test could be that hidden variables cannot restore locality (a conclusion preferred by John S. Bell and others). In other

words, a hidden-variable description of experimental phenomena must forgo the principle of local causality. However, it doesn't imply that there is no quantum-mechanical description compatible with locality.

If the authors agree with me, I hope the authors can clarify the relation summarized in Eq. (11) of the SI. In particular, I hope the authors can emphasize that a hidden-variable description is as important as the other two assumptions (locality and measurement independence) in order to derive a Bell inequality. A similar clarification can be made for the relation summarized in Eq. (12) of the SI.

We thank the Referee for their remark. We agree that the conclusion of a loophole-free Bell test (with space-like separation and measurement independence) implies that hidden variables cannot satisfy local causality. However, the same conclusion holds for models with no hidden variables as well. Indeed, hidden variable models are not a restriction on the set of models, but it is the other way around: models without hidden variables are a special case of hidden variable models for which the quantity λ takes a unique value. Therefore, any conclusion valid for hidden variable models also applies to the special case of models without hidden variables. This is the reason why Eq. (1) can be defined “without loss of generality”.

The case of a quantum description with no additional variable is in fact a clear example in which a model violates the locality condition Eq. (2). For σ_z measurements performed on a ϕ^+ Bell state, setting $\lambda = |\psi\rangle$, the model yields $P(+1|a,b,+1,\lambda) = 1$ while $P(+1|a,\lambda) = 1/2$, hence clearly violating Eq. (2). As discussed in the last paragraph of Appendix I, violation of Eq. (2) does not imply signaling, and indeed one can check that this model satisfies the no-signaling conditions Eq. (13).

Since the hidden variable decomposition Eq. (1) is not an assumption, i.e. a restriction on the considered models, but rather a freedom that is given to the models, it is not required on the lhs of Eqs (11) and (12). The relations expressed there are true even when this freedom is not used. The fact that the assumptions listed on the left hand side of these equations are sufficient to infer the right hand side is rigorously proven by the mathematical steps provided in Appendix A.

2) To obtain Eq. (5) in Section I of the SI, it is mentioned in lines 89-90 that Bayes' rule is used. Instead of Bayes' rule, I think the chain rule in probability theory should be used.

Following the suggestion of the Referee, we have changed “Bayes' rule” to “the chain rule of probability theory”.

3) In lines 86, 87, 88, 101, 102, and 108 of the SI, the distribution of the hidden variable λ is denoted by the little-case letter p , which is not consistent with the notation introduced in line 77.

We thank the Referee for pointing out this notation inconsistency. The lowercase letters p have been replaced by uppercase letters P .

4) The p-value is defined in the first sentence (lines 1218-1220) of Section X.B in the SI. I think this definition could be more precise. There are many different local hidden variable models, and the probabilities according to different models of producing statistics at least as extreme as the experimental one can be different. Given these, one can figure out the best local hidden variable model according to which the probability of producing a statistic at least as extreme as the experimental one is the highest. Such the highest probability is the p-value for testing local hidden variable models.

In the resubmitted manuscript we have changed the first sentence in SI Section X.B as follows: “The p-value of the Bell test is given by the probability with which any local hidden variable model could reproduce statistics at least as extreme as the results of our experiment.”

5) About the success probability p_{win} in Eq. (19) of Section X.B in the SI, it is not clear from that equation whether the success probability is according to a local hidden variable model or according to a quantum strategy. I suggest the authors clearly define the success probabilities according to different strategies. For example, the authors could introduce two variables $p_{\text{win}}^{\text{LHV}}$ and $p_{\text{win}}^{\text{QM}}$ in the first paragraph of Section X.B. Then it would become clear which success probability should be referred to in the equations below that paragraph.

We agree that the suggested change makes sense and we have adapted SI Section X.B accordingly.

6) In lines 1326-1327 of the newly added Section X.C, it is mentioned that a_i (b_i) and x_i (y_i) are the outcome and setting of Alice (Bob) observed in round i of the experiment. The notation here is not consistent with that used elsewhere in the manuscript: In this work, a and b are used to denote the settings of Alice and Bob, and x and y are used to denote their outcomes. I also find that the term "round" has been used several times in Section X.C, while elsewhere in the manuscript the term "trial" has been used.

We thank the referee for pointing out these inconsistencies, which we have corrected in the resubmitted manuscript.

Referee #3 (Remarks to the Author):

The authors have implemented significant updates to the manuscript in response to the first round of revision requests. I have a few additional comments, some about elements in the original manuscript that have become more clear with the revision, and some about the revised sections. Some of these may seem like language nits, however for this topic in particular it is valuable to use language with careful intention.

Main text

Line 65: “early experiments relied on additional assumptions, creating loopholes in Bell’s argument.”

>> No, Bell's argument is sound. It is the mapping of a gedankenexperiment into the lab that introduces the assumptions and thus the loopholes.

We thank the Referee for this remark. We agree that the loophole does not reside in Bell's argument but in the early experimental realizations, when a number of unnecessary assumptions were used to conclude that the reported results were incompatible with an explanation satisfying the principle of local causality. We have adapted the sentence accordingly.

Line 71: "As quantum technology matured..." this suggests that quantum technology has matured; arguably it's still pretty nascent. Consider alternate verbiage, e.g. "In the development of quantum information science, ..."

We have changed the text according to the suggestion of the referee.

Line 80: "(the choice of measurement)" The word "measurement" is especially ambiguous when it gets close to quantum mechanics—is this the choice of what to measure, how to measure, when to measure, whether to measure...? Consider being more explicit: "(the choice of measurement basis)".

We have changed the corresponding sentence accordingly.

Line 108-111: "While these experiments all relied on additional assumptions, in this work, we set out to demonstrate a loophole-free violation of Bell's inequality using superconducting circuits."

>> In my eyes, stating which loopholes an experiment claims to close is important to the overall claim of the paper. I could not find a place in the manuscript where it is articulated which loopholes will be addressed. No experiment is truly "loophole-free", and the term is vague enough that a case-specific description is warranted—this seems like a great spot to add it. (Yes, there is a citation of (Larsson, 2014), however this reference includes mention of a number of loopholes including some that aren't actually relevant to this setup. Regardless of the citation, any strong manuscript should describe clearly at least once which loopholes it claims to close.)

In the resubmitted manuscript we have added an explicit statement in lines 161-165, as suggested by Referee 3 in a separate comment: "In the following we discuss how we fulfill the requirements outlined here for realizing a Bell test with superconducting circuits closing the locality, fair-sampling and memory loopholes and supporting measurement independence all at the same time."

Line 114: The "locality loophole" is introduced without explanation in the main text. Its definition is central to the manuscript's claims. Consider inserting a reference to the Supplementary information where the loophole definition is more clearly explained. (This is also relevant in line 368.)

We consider the whole paragraph that follows line 114 as an explanation of the locality loophole. In addition, in the revised manuscript we have added a new paragraph at the end of Supplementary

Section I with explicit definitions of the loopholes and we refer to that Section at the corresponding location (line 118) in the main text.

Line 119: To appeal to a broad audience, consider defining “concurrency”

The concurrency is an entanglement monotone depending on the eigenvalues of the a particular Hermitian matrix that is built from the density matrix of the Bell state. Unfortunately, this mathematical construct is not very intuitive per se, and we believe that defining it at the suggested place in the manuscript would require more detailed mathematical explanations, which in turn would not appeal to a broad audience. For this reason we prefer to keep the description of the concurrency brief and refer to external references for further information. In order to appeal better to a broad audience, we have made the intuitive meaning more clear in the revised version by adding “(...) where C [40,41] is a measure of the degree of entanglement present in the system.”

Line 132-4: Consider the following adjustment for clarity/simplicity: “the corresponding measurement outcomes by the party at one site are unknown to the party at the other site”

We have implemented the suggested change in the revised manuscript.

Line 135: “The locality condition (Supplementary Section I) is also commonly considered as a fundamental ingredient of a Bell test rather than a loophole [3, 19].”

>> My impression: this sentence almost seems to undermine the entire section, it’s not clear to me that it adds strength or clarity to the claims. Consider removing it. If you must keep it, consider relocating it to the Supplement.

We agree with the referee that this sentence is not necessary and we have therefore removed it.

Line 145: Like the comment in Line 114, the “detection loophole” is not defined; unlike the comment in 114 I can’t find a definition in the supplement. I suspect the authors actually want to refer to the “fair-sampling” loophole; the term “detection loophole” is rather informal, and originated in photon experiments where a fair-sampling assumption may come as a result of low photon detection rates. Consider updating and defining (somewhere) the loophole term.

We agree that the equivalent term “fair-sampling” is more general and thus matches better for our experiment. We have therefore replaced the term “detection loophole” by “fair-sampling loophole” in total 5 times in the main text, methods and supplementary information. In a new paragraph in Supplementary Section I, which we refer to in the main text at the appropriate places, we give a brief, explicit definition of the loopholes in the revised manuscript.

Line 156-7: “If the properties of the two entangled qubits were described by a local hidden variable model”

>> For strength of argument, consider “If the properties of the system were described...” The goal of a Bell experiment is to test a system’s ability to be described by hypothetical LHVs; we need not assume that said system is quantum in order for the test to be valid.

We have adapted this sentence accordingly in the revised manuscript.

Line 162-163: “realizing a Bell test closing all major loopholes” >> this would be another great place to reference the specific list of loopholes the experiment aims to close.

We have rewritten this sentence in a more explicit way: “In the following we discuss how we fulfill the requirements outlined here for realizing a Bell test with superconducting circuits closing the locality, fair-sampling and memory loopholes and supporting measurement independence all at the same time.”

Line 184: nit. “which” → “that”

We have changed the word accordingly.

Line 187: Please remove “to assert non-locality”. The experiment doesn’t “assert non-locality”, rather it leverages space-like separation to address the locality loophole.

We have adapted the sentence as follows: “In a Bell test that closes the locality loophole, minimizing the duration of the readout reduces the distance d required between the two parties to provide space-like separation.”

Line 212: For clarity, consider “due to” → “by leveraging”. (I had to read this sentence a few times before I understood it.)

We have implemented the suggested change.

Line 223: “major cryogenic system” I’m puzzled by the word “major”; Is this well-defined jargon? Maybe “large-scale”?

As suggested by Referee 3, we have replaced “major” by “large-scale”.

Lines 267-269: “We consider the start event of each trial of a Bell test as being marked in space and time by the creation of a random number in each RNG.”

>> Each trial should have a single start event; here two are mentioned (RNGA and RNGB).

We have adjusted the phrasing to make this aspect more clear in the revised manuscript: “We consider the start event of each trial of a Bell test as being marked in space and time by the earlier of the two events marking the creation of a random number in each RNG.”

Lines 273-275: “a and b become available as voltage pulses at the output of the RNGs 17.10 ± 0.14 ns after this event”

>> Perhaps connected to the “multiple start events” confusion from lines 267-269, I got a little confused by this sentence. Consider making this sentence a single-RNG sentence, rather than a general statement about the overall experiment, e.g. “Each setting output from a RNG (a and b respectively) becomes available as a voltage pulse at the RNG’s output 17.10 ± 0.14 ns after the random number’s creation.”

In the revised manuscript we have rephrased this section in terms of a single node, as suggested by Referee 3.

Lines 280-287: I struggled to match the text here with the corresponding blocks in figure 4, in part because figure 4 contains five separate blocks that could be interpreted as close to “dark green”, and in part because the text in these lines doesn’t match word-for-word with the figure labels. For me, establishing matching words between the manuscript text and the figure would be enough.

In order to improve clarity here, we now refer to the segment in Figure 4 as “turquoise” rather than “dark green” and we explicitly refer to the propagation delay as the “signal propagation delay”, matching the label in Figure 4.

Line 299: For clarity, consider “As for” → “As done with”. (I had to read this sentence a few times before I understood it.)

We have implemented the suggested change in the revised manuscript.

Line 301-2: For clarity, consider “In this way we minimize the propagation delay... to 14ns...” → “In this way we minimize to 14 ns the propagation delay...”. (I had to read this sentence a few times before I understood it.)

We have changed the sentence as suggested in the new version.

Lines 417-425: This seems to mostly repeat the prior paragraph, feeling a little rambly. Consider distilling the new material into a single DI-focused sentence and cutting the rest.

We thank the referee for this comment. In our opinion, lines 417-425 form an important conclusion of our work. After having discussed the S-value and the repetition rate individually, we here proceed to clarify that it is the combination of both which matters for practical implementations of device-independent quantum information processing protocols, and that our setup enables such experiments because of an interesting combination of the aforementioned metrics. We therefore prefer to keep this paragraph as it is.

Lines 430-433: “Interconnected cryogenic systems may indicate a pathway towards realizing larger scale quantum computing systems using quantum microwave local area networks.”

>> This is very surprising to me—naively I’d expect this type of setup to be prohibitively challenging to deploy at scale! Perhaps the authors have learned, through building this experiment, about some practical challenges one would face to deployment at scale, or what makes this easier than expected. If so, that would be very interesting to hear.

We agree with the Referee that this approach is technically challenging. However, as our experiment demonstrates, it can be done successfully, especially now that the initial engineering and design work has been completed. As we discuss in Supplementary Information, Section VIII.B, such a setup could also be extended to larger distances of more complex geometries in a straight-forward manner. Please note that we plan to discuss these technical aspects of the cryogenic system, including aspects of scaling up, in a separate publication. Furthermore, we envision that the approach presented in this publication could be used to interconnect several cryogenically cooled quantum computing systems within a quantum computing center (on distance scales of tens of meters). To clarify this, we have changed the sentence in lines 443-447 as follows: Interconnected cryogenic systems may indicate a pathway towards realizing larger scale quantum computing systems using quantum microwave local area networks [51], e.g. within a quantum computing center.

Line 517: “withe” ?

We have corrected this typo in the revised version.

Line 518, Supplement section IV “Quantum Devices”: Consider adding just one more explanatory sentence in at least one of these locations to explain why both “qubit” and “qutrit” appear in the manuscript and how a reader should interpret when they see one vs the other—otherwise the switch (paper mostly uses qubit, supplement mostly uses qutrit) is a little disorienting.

We have added a clarifying sentence in the Methods section “Superconducting Qubits” (lines 763-765): “Here, as well as elsewhere in this publication, we refer to the quantum bit as a qutrit when we refer to its lowest three energy eigenstates.”

Lines 609-612, 613-641: “This consideration has no relevance to our situation, because we use a deterministic entanglement generation protocol which is independent of the outcome of any measurement.”

>> The section in 613-641 is long, verbose, and slightly dismissive; another (simpler) way to articulate this is that for a given assumption about setting choice origin (which is made already in order to address the locality loophole) there are certain possible origins of λ that can be excluded by the experiment due to space-time layout, and other possible origins of λ that cannot be excluded through space-time layout. This statement affects the present experiment just like any other. Thus to suggest that “This consideration has no relevance to our situation” seems to miss the point. If the goal is to address ref#2’s request, consider an actual space-time comparison; at least please adopt a less dismissive tone.

We thank the Referee for this comment and we take the criticism seriously. It was certainly not our intent to be dismissive, rather our aim was to provide an explanation of the role that space-time separation plays in closing loopholes in the main Bell test scenarios.

In particular, we described two scenarios other than our own: event-ready Bell tests in which a heralding event signals that entanglement has been generated, and photonic Bell tests in which a photon-pair generation event is associated with the creation of entanglement. When we wrote “This consideration has no relevance to our situation...” we were referring to event-ready Bell tests: since there is no heralding event in our implementation, this consideration is simply not applicable to our experiment. The Referee seems to have misunderstood the scope of our “no relevance” claim. In order to avoid any possible misunderstanding, we made the following change in the revised version: “Since we use a deterministic entanglement generation protocol which is not dependent of the outcome of any measurement, such considerations related to a heralding-event do not play a role in our experiment.”

Other than this clarification, we believe it would be inappropriate at this late stage of the review process to alter the current text, which was explicitly requested by Referee 2, and has been approved by Referee 2 in its current form.

Supplement

Lines 123-124: “For instance, in a space-time configuration allowing for communication from A to B,”
>> Since A and B define spatial locations only, any configuration allows for such communication.

We have clarified this by changing the sentence as follows: “(...) allowing for communication from A to B during a trial of the Bell test, (...)”

Lines 133-135: "On the other hand, a theory that is not locally causal will be referred to as non-local".

>> Per previous discussion of the term "non-local", this could be misleading. The definition also is not used elsewhere. Please remove this sentence.

The terms "non-local" and "non-locality" are used in several places of the main text, including in the abstract and in the outlook. We think it is important to define their meaning within the context of our work. This is the role of this short remark.

Lines 136-138: "In order to test locally causal theories, we are interested in configurations where their statistics admit a local hidden variable model."

>> I don't understand what this sentence is supposed to communicate, in particular the phrase "their statistics".

We changed "their statistics" for "their accumulated statistics", which is a terminology that is already used in the main text.

Lines 162-165: "In particular, this conclusion requires an appropriate definition of the events at which the inputs a and b are created and those at which the outcomes x and y are fixed."

>> Consider a cleaner phrasing: "In particular, this conclusion relies on an assumption about the space-time points at which a and b are created and at which x and y are fixed."

We are of the opinion that the phrasing is already clear when one also considers the following sentence.

Lines 165-168: "Still, this shows that under the principle of local causality, the locality condition (2) reduces to an assumption on the time and location at which events happen."

>> This phrasing is problematic (esp "reduces to"), and I don't see that this sentence adds anything. Consider removing it.

We think this sentence and paragraph is critical, and that it should therefore remain in the paper. We have however replaced the term "reduces to" with "transforms to".

Lines 185-186: "The amount of measurement independence needed to conclude can however..."

>> Needed to conclude what?

We have clarified this sentence as follows: "The degree of measurement independence between the choice of measurement settings and the tested local hidden variable models can however be arbitrarily small [5]."

Lines 200-208: The authors state that “realism” is not needed and therefore their conclusion is clearer. I actually found the discussion leading up to this statement was less clear than other parts of the manuscript, so would shy away from emphasizing clarity here. I don’t understand what the authors refer to under “realism”, and thus I don’t understand what exactly they claim to avoid. For example, some might define realism as “LHVs exist”--which is an inevitable component of discussing hypothetical LHV models. I’d encourage the authors to clarify what they mean by realism, or better yet, to increase clarity, remove the section entirely. In my eyes, this section weakens, not strengthens, the manuscript.

Following the Referee’s comment, we toned down this section by removing the last sentence, hence removing any claim of increased clarity. While our derivation does not involve a concept of realism, other derivations did, and so we believe that it is helpful to the reader to point this out as a remark. As described in Ref. [11], these two approaches are not contradictory, but complementary: both are valid. It is the purpose of this paragraph to point this out. Describing the details of all possible derivations of Bell’s inequalities is out of scope here. Other works discussed these aspects in detail and more information on this topic is available for the interested reader in the cited literature.

Line 213: “fact” → “assertion”

We agree that the word “fact” could be misleading in this context. We have changed it to “condition” which we find even more clear than “assertion”.

Line 409: remove “non-local”

We prefer to keep “non-local” here, as the main reason for building a 30-meter long cryogenic setup was to close the locality loophole.

Line 544: “to at” >> pick one

We have corrected that typo and solely kept the word “to”.

Fig S5: The updated figure is so much easier to understand. Thank you for the modifications, I find them very helpful.